



# Soil Parameterization in Land Surface Models Drives Large Discrepancies in Soil Moisture Predictions Across Hydrologically Complex regions of the Contiguous United States

Kachinga Silwimba[1], Alejandro N. Flores[1], Irene Cionni[1], Sharon A. Billings[2], Pamela L. Sullivan[3], Hoori Ajami[4], Daniel R. Hirmas[5], and Li Li[6]

[1]Department of Geosciences, Boise State University, Boise, ID, USA
[2]Department of Ecology and Evolutionary Biology and Kansas Biological Survey & Center for Ecological Research, University of Kansas, Lawrence, KS, USA
[3]College of Earth, Ocean, and Atmospheric Science, Oregon State University, Corvallis, OR, USA
[4]Department of Environmental Sciences, University of California Riverside, Riverside, CA, USA
[5]Department of Plant and Soil Science, Texas Tech University, Lubbock, TX, USA
[6]Department of Civil and Environmental Engineering, Pennsylvania State University, University Park, PA, USA

**Correspondence:** Kachinga Silwimba (kachingasilwimba@u.boisestate.edu)

**Abstract.** Land surface models (LSMs) are critical components of Earth system models (ESMs), enabling simulations of energy and water fluxes essential for understanding climate systems. Soil hydraulic parameters, derived using pedotransfer functions (PTFs), are key to modeling soil-plant-water interactions but introduce uncertainties in soil moisture predictions. However, a key knowledge gap exists in understanding how specific soil hydraulic properties contribute to these uncertainties

and in identifying the regions most affected by them. This study assesses the influence of soil parameter settings on soil moisture variability in the Community Land Model version 5 (CLM5) over the contiguous United States (CONUS) using Empirical Orthogonal Function (EOF) analysis. EOF analysis identified dominant spatial and temporal soil moisture patterns across multiple experimental configurations and highlighted the impact of soil parameter variability on hydrological processes. The results revealed significant discrepancies in soil moisture simulations, particularly in the central Great Plains, potentially

due to the combination of arid climate conditions and limitations in modeling saturated hydraulic conductivity and soil water retention curves. Seasonal soil moisture dynamics aligned broadly with observed patterns but showed biases in magnitude and phase, emphasizing the need for refined parameterization, such as improving the representation of infiltration and drainage processes. Comparisons with ERA5-Land reanalysis data revealed improved alignment in regions with consistent climatic gradients but persistent model deficiencies in hydrologically complex areas, particularly under more arid climates such as the

Great Plains where hydrological processes are notoriously harder to reproduce. Ths research highlights the necessity of refining soil parameter representations, utilizing high-resolution datasets, and considering climatic variability to boost the performance of LSMs. Importantly, these findings also open the door to future efforts that incorporate dynamic soil properties into LSMs. Much of this work demonstrates the dynamism of soil properties, and while this study advances modeling by revealing the importance of their inclusion, the next crucial step will be developing approaches that allow these properties to be dynamic



within LSMs. This paper serves as a foundational step toward that goal, paving the way for more complex and integrated modeling frameworks that better capture soil-hydrology-climate interactions.

## 1    Introduction

Land surface models (LSMs) are essential components of Earth system models (ESMs), offering critical insights into the movement and partitioning of energy and water across the Earth's surface, which are fundamental processes in understanding

and simulating climate systems accurately (Kang and Hong, 2008; Zhao et al., 2017; Guimberteau et al., 2017; Hagemann et al., 2013; Dagon et al., 2020). Designed to operate on large spatial scales, LSMs rely on robust parameterizations of land processes, including the use of pedotransfer functions (PTFs) to parameterize soil hydraulic properties. PTFs, as described by Van Looy et al. (2017) and De Lannoy et al. (2014), are mathematical formulations that use extensive soil hydraulic databases to establish empirical relationships between soil particle-size distribution and soil hydraulic parameters, such as

field capacity, permanent wilting point, saturated hydraulic conductivity, pore-size distribution, and soil water retention curves (McNeill et al., 2018; Vereecken et al., 2010; Weber et al., 2020). These PTFs range in complexity from basic linear models to advanced machine learning algorithms such as artificial neural networks. These soil hydraulic parameters are fundamental to quantification of soil moisture and water flow, and soil-plant-water interactions and their effects on climate, agriculture, hydrology, and environmental engineering.

PTFs play a crucial role in converting readily available soil texture data into soil hydraulic parameters, addressing the difficulties of acquiring accurate soil moisture data at larger scales (Fu et al., 2023). However, many soil hydraulic parameters are derived from laboratory or small-scale field studies, which often fail to capture the full heterogeneity of larger areas, limiting their representativeness (Lai and Ren, 2016; Godoy et al., 2018). To overcome this limitation, global soil texture maps enhance PTFs' predictive capabilities, enabling their application in regions where field measurements are unavailable and making them

indispensable for land modeling (Tafasca et al., 2020; Dai et al., 2019). Soil moisture, a key output of these models, is a vital variable governing the exchange of water and energy between land and atmosphere. It has profound impacts on climate systems, vegetation dynamics, and extreme events, including droughts and floods (Zhang et al., 2021).

The influence of soil hydraulic properties on soil moisture simulations is well documented. For example, Fu et al. (2023) demonstrated that these properties significantly affect soil moisture simulations at the ELBARA field site in the northeast of

the Tibetan Plateau, using the one-dimensional (1D) Richards equation. Similarly, Fu et al. (2022) noted that the numerical solution approach of the Community Land Model (Lawrence et al., 2019) produces a narrow range of soil hydraulic property values, which suggests a relatively weak influence on soil moisture simulations within this range. However, when optimized hydraulic properties are used potentially derived to capture site-specific variability or improve model performance beyond this narrow range they can exert a more substantial influence on soil moisture dynamics. Furthermore, Feki et al. (2018)

highlighted that saturated hydraulic conductivity exhibits the highest sensitivity to temporal changes in environmental factors, such as precipitation or temperature variability significantly affecting soil moisture variability, as shown in FEST-WB model simulation of a maize field in the Secugnago region. These findings underscore the importance of accurately representing soil





hydraulic properties, which directly influence the partitioning of water into runoff, infiltration, and evapotranspiration (Ye et al., 2023), as well as the temporal and spatial variability of soil moisture. However, uncertainties in parameterizations, such as the
soil water retention curve that links water potential to volumetric soil moisture, continue to challenge the predictive capacity of LSMs, especially under extreme climatic conditions (Koster et al., 2004; De Lannoy et al., 2014). Improving the representation of soil moisture and its underlying hydraulic properties is critical, as it affects global hydrological cycles, vegetation health, and energy flows, all of which are essential for understanding and mitigating the impacts of climate change (Oleson et al., 2010).

In addition to these complexities, scaling point-scale or regional observations of soil moisture to the coarser resolutions of LSM outputs presents a persistent challenge. While observational networks and remote sensing missions have expanded the availability of soil moisture data, the heterogeneous nature of soil properties combined with varying retrieval algorithms and coverage gaps can introduce significant uncertainties, both in terms of the accuracy of satellite products and their limitations for validating LSM outputs (Famiglietti, 2014; Brocca et al., 2017). Moreover, uncertainties in parameterization make it challeng-
ing to accurately simulate soil moisture dynamics, as noted by Reichle et al. (2004) and Kato et al. (2007), limiting the ability of LSMs to replicate observed soil moisture datasets. This discrepancy in spatial resolution and data precision can make model calibration more challenging, increase uncertainties in estimating parameters, and, as a result, weaken confidence in simulation outputs. Emerging evidence further complicates this issue by highlighting that soil properties can change over relatively short time scales due to shifts in climate and land cover. The dynamic nature of soil properties introduces additional pressure to
better understand soil-hydraulic relationships and integrate these temporal dynamics into land surface models, as demonstrated by studies highlighting how climate and land cover changes influence soil processes (Hirmas et al., 2018; Koop et al., 2023; Caplan et al., 2019; Sullivan et al., 2022; Hauser et al., 2022). Addressing these complexities demands robust, data-oriented approaches and dimensionality reduction techniques to disentangle the effects of parameterization on soil moisture patterns across ecosystems and climate conditions.

A major challenge to addressing these uncertainties is the high dimensionality of LSM simulations when applied to continental or global scales, making it difficult to isolate the effects of specific parameters on soil moisture from other factors such as meteorological forcings and modes of climate variability Ji et al. (2023); Li et al. (2013); Zeng et al. (2021). This research investigates two critical questions: (1) How do soil hydraulic parameters influence large-scale spatial patterns in soil moisture associated with well-characterized climate variability modes? (2) How do these parameters affect the temporal dynamics of
soil moisture during climate extremes, such as droughts and floods? Using EOF analysis, the study systematically evaluates the impact of soil hydraulic parameterizations in CLM5 simulations in the contiguous United States (CONUS). This study enhances comprehension of soil-plant-water dynamics by isolating parameter effects, thereby improving predictions of eco-hydrologic responses to climate variability and change, tackling a crucial challenge in land modeling and climate forecasting. We elaborate on the methodologies employed in Empirical Orthogonal Function (EOF) analysis, covering data sources and
computational methods, and present the principal findings derived from the CLM5 simulations, highlighting their relevance to soil moisture variability and parameter sensitivity. Additionally, the sections discuss the broader impact of these findings on



the advancement of land surface modeling and the comprehension of climate dynamics. Finally, they conclude with practical recommendations for upcoming research and applications in the fields of ecohydrology and climate science.

## 2 Data and Methods

### 2.1 Study Region

The study area depicted in Figure 1 covers the CONUS, which extends from the Atlantic to the Pacific Ocean and is bordered by Canada to the north and Mexico to the south. This region features a diverse range of climate zones, including humid continental, Mediterranean, subtropical, arid, and alpine, all shaped by its extensive geographic reach and varied topography. Such climatic variance plays a crucial role in influencing soil moisture patterns, leading to dynamic variations throughout CONUS. The region's topographical elements, like the Rocky Mountains, Appalachian Mountains, Sierra Nevada, and Cascade Range, significantly affect precipitation patterns, runoff, and soil characteristics, further contributing to soil moisture variability. Additionally, land cover types vary significantly, with moisture-rich forested areas and urban regions with limited infiltration capacity introducing further complexity to soil moisture distribution. To improve the analysis of climate, topography, and land cover interactions on soil moisture dynamics, this study employs the regional subdivisions of CONUS suggested by Giorgi and Francisco (2000). As illustrated in Figure 1, the area is divided into four primary climate zones: Western North America (WNA), Central North America (CNA), Eastern North America (ENA), and North Central America (NCA). These subdivisions enable a more detailed examination of soil moisture variability within distinct climatic and geographic contexts, providing valuable insights into the processes affecting soil moisture dynamics.



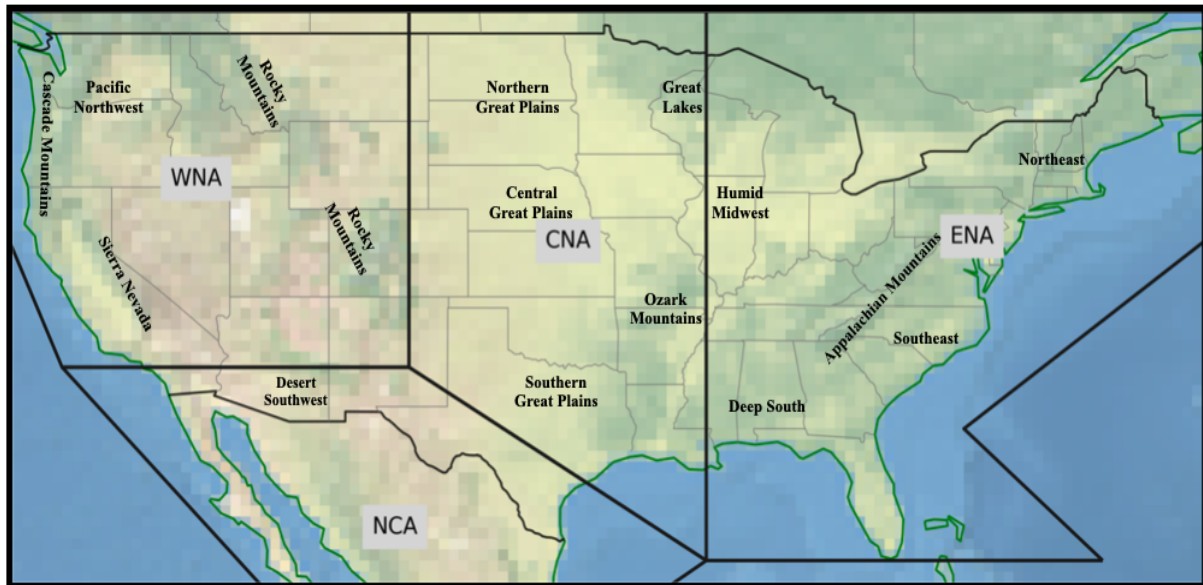

**Figure 1.** Regional divisions of the CONUS area into four major zones: Western North America (WNA), Central North America (CNA), Eastern North America (ENA), and North Central America (NCA), as defined by Giorgi and Francisco (2000), based on climate variability and geographical features. Prominent subregions and geographical landmarks, such as mountain ranges and plains, are also depicted.

## 2.2 Data Description

The Soil Parameter Intercomparison Project (SP-MIP), initiated at the GEWEX-SoilWat workshop in Leipzig (2016), aims to quantify the variability in land surface model (LSM) output caused by differences in soil parameters and structures. Following the Land Surface, Snow, and Soil Moisture Model Intercomparison Project (LS3MIP) protocol (Van den Hurk et al., 2016), SP-MIP brought together eight leading climate land models CLM5, ISBA, JSBACH, JULES, MATSIRO, MATSIRO-GW, NOAH-MP, and ORCHIDEE for a series of global simulation experiments (Gundmundsson and Cuntz, 2017). These models

were run on a $0.5°$ grid using Global Soil Wetness Project Phase 3 (GSWP3) meteorological forcing data for 1980 to 2010. Four experimental designs were implemented to isolate the effects of soil properties on hydrological and energy balance variables. Soil parameters for Experiment 1 and soil textures for Experiment 2 (EXP2) were derived at a $0.5°$ resolution, based on dominant soil classifications within the 0-5 cm layer of SoilGrids data (Hengl et al., 2014) at a 5 km resolution. The Brooks and Corey parameters are derived from Table 1 of Clapp and Hornberger (1978), while the Mualem-van Genuchten

parameters represent ROSETTA class average hydraulic values as cited by Schaap et al. (2001), with soil textures taken from Table 1 of Cosby et al. (1984). For Experiments 4a-d (EXP4a–4d), the USDA soil categories used are Loamy Sand, Loam, Silt, and Clay, as defined by Montzka et al. (2011), employing identical transfer functions for Brooks and Corey and Mualem-van Genuchten parameters as applied in Experiment 1 (EXP1). All models are assumed to solve the Richards equation for soil water movement. The provided soil parameters and textures are uniform throughout the entire soil column. For a detailed description

of the SP-MIP dataset, please refer to (Gundmundsson and Cuntz, 2017).



This study uses soil moisture data from the CLM5 experiments developed by the National Center for Atmospheric Research (NCAR) (Thornton, 2010; Lawrence et al., 2019). The schematic (Figure 2) illustrates the CLM5 modeling framework, depicting the experimental setup for seven different model runs, each designed to evaluate the influence of soil hydraulic parameterizations on soil moisture variability. The dataset covers global landmasses at $0.5°$ resolution (25,920 grid cells, excluding
water bodies and permanent snow/ice) and includes 41 land surface variables such as evapotranspiration, soil temperature, and runoff, spanning 30 years (1980 to 2010). The global soil profile reaches a depth of 41.998 m with 25 layers, but for this study, soil moisture was extracted from depths (0-1.0 m) containing most roots (root zone) of the CONUS region, covering 6,413 grid cells. The focus is on the variable water content of soil layers (`mrsol`) to explore soil moisture variability and distribution.

### 2.2.1 Experimental Designs

(1) EXP1: A baseline control was conducted using globally standardized soil hydraulic parameters from SP-MIP (refer to Table 1) to examine the presumed decrease in inter-model variability when uniform soil parameters are employed.

(2) EXP2: Soil texture properties provided by SP-MIP (see Table 2) utilized lookup tables or PFTs to determine hydraulic parameters, analyzing the variability caused by parameter translation methods.

(3) EXP3:The model applied its default soil hydraulic settings, maintaining consistent soil properties throughout all layers
to explore the intrinsic inter-model variability.

(4) EXP4a–4d: These experiments utilized key soil hydraulic parameters sourced from SP-MIP (refer to Table 1), employing uniform parameters across four different soil categories: loam, clay, silt, and loamy sand. This configuration explored how sensitive LSMs are to soil hydraulic parameters and the impact on spatial variability in both hydrological and energy balance outputs.



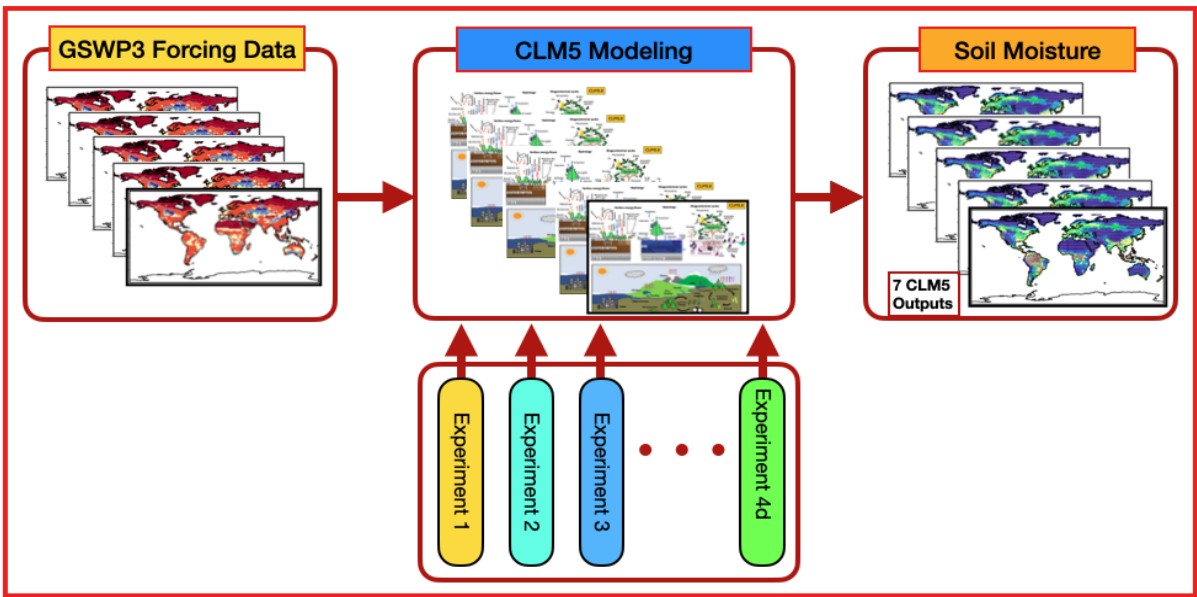

**Figure 2.** Experimental setup for evaluating soil moisture variability in CLM5. The model uses GSWP3 forcing data and runs multiple experiments with different soil hydraulic parameterizations. EXP1 applies standardized parameters, EXP2 derives parameters from soil texture, EXP3 uses default CLM5 settings, and EXP4a–4d assign uniform parameters for different soil types.

### 2.2.2 Benchmark Dataset

The ERA5-Land dataset, provided by the European Centre for Medium-Range Weather Forecasts (ECMWF), serves as a key reference for model evaluation. Unlike other models, ERA5-Land does not directly incorporate soil moisture observations. Instead, it uses atmospheric data from the ERA5 reanalysis, which integrates meteorological and satellite observations via a 4-D variational assimilation system coupled with a simplified extended Kalman filter (Muñoz-Sabater et al., 2021). This methodology enables land surface changes to be primarily guided by modeled processes while being affected by larger atmospheric conditions. In terms of soil moisture, the ERA5 system assimilates information from a range of satellite sources, such as the Soil Moisture Ocean Salinity (SMOS), Advanced Microwave Scanning Radiometer-2 (AMSR-2), Tropical Rainfall Measuring Mission Microwave Imager (TRMM-MI), ERS-1/2 Active Microwave Instrument scatterometer, and Meteorological Operational Satellite (De Rosnay et al., 2013). Although ERA5-Land uses an indirect method for assimilation, it is often employed as a reference for validating soil moisture data due to its global consistency and frequent updates. However, studies have pointed out certain discrepancies, like a wet bias in its soil moisture measurements relative to ground-based and SMAP satellite data, especially in heavily vegetated and humid areas (Lal et al., 2022). These biases highlight the importance of careful interpretation when applying ERA5-Land to hydrological tasks. Despite these issues, its capacity to reflect broad spatiotemporal patterns ensures its effectiveness in assessing model performance and conducting extensive hydrological research. While alternative datasets such as the North American Land Data Assimilation System (NLDAS) could provide higher resolution



and are region-specific to CONUS, ERA5-Land was selected for its global consistency, frequent updates, and ability to offer a broader perspective that facilitates comparison across varying climatic conditions. Additionally, ERA5-Land provides a direct connection to global atmospheric reanalysis, enabling robust assessments of large-scale interactions between soil moisture and climate processes. The ERA5-Land dataset was regridded to fit the CLM5 $0.5°$ resolution.

**Table 1.** Soil parameters for the three selected water retention curves were supplied by SP-MIP as input for experiments 1 and 4a-d.

| Parameter Name | long_name (netCDF) | Unit |
| --- | --- | --- |
| he | air entry potential | m |
| mbc | Brooks-Corey m parameter = Clapp-Hornberger b | – |
| thetar | residual soil moisture | $m^3\ m^{-3}$ |
| thetas | saturated soil moisture, porosity | $m^3\ m^{-3}$ |
| ks | Hydraulic conductivity at saturation or at air entry | $ms^{-1}$ |
| lambdac | Corey lambda parameter | – |
| alphavg | van Genuchten alpha parameter | $m^{-1}$ |
| nvg | van Genuchten n parameter | – |
| mvg | van Genuchten m parameter | – |
| thetafcbc | Brooks-Corey field capacity | $m^3\ m^{-3}$ |
| thetafcvg | van Genuchten field capacity | $m^3\ m^{-3}$ |
| thetapwpbc | Brooks-Corey permanent wilting point | $m^3\ m^{-3}$ |
| thetapwpvg | van Genuchten permanent wilting point | $m^3\ m^{-3}$ |

**Table 2.** Soil textural characteristics supplied by SP-MIP for experiment 2.

| Parameter Name | long_name (netCDF) | Unit |
| --- | --- | --- |
| fclay | fraction of clay | – |
| fsilt | fraction of silt | – |
| fsand | fraction of sand | – |
| rhosoil | dry bulk density | $kgm^{-3}$ |
| omsoil | organic matter content | $g(C)kg^{-1}$ |

## 2.3   EOF Analysis for Soil Moisture Variability

EOF analysis is a well-established statistical technique used in geophysical sciences to reduce the dimensionality of complex datasets while preserving their most significant variability patterns (Jollife, 2002). Originally introduced in meteorology by Lorenz (1956), it has been widely applied to analyze spatio-temporal variability in climate and hydrological systems (Monahan





et al., 2009). EOF analysis systematically decomposes large datasets into orthogonal spatial patterns (EOF modes) and their
corresponding temporal components (principal components, PCs), enabling the identification of dominant variability modes
and their temporal evolution. In this study, EOF analysis is applied to soil moisture fields simulated by CLM5 across the
CONUS region. This approach isolates spatial and temporal patterns of soil moisture variability, providing insights into the
influence of soil hydraulic parameterizations on large-scale moisture distribution and the temporal dynamics of extremes such
as droughts and floods. Previous studies have demonstrated the effectiveness of EOF in linking soil moisture variability to
large-scale climate drivers, such as the El Niño-Southern Oscillation (ENSO) and drought-flood cycles (Bauer-Marschallinger
et al., 2013; Korres et al., 2010), highlighting its utility in disentangling the role of parameterization from other drivers (Lorenz,
1956; Jawson and Niemann, 2007). The EOF modes extracted in this study capture dominant spatial patterns, while the PCs
characterize their evolution under varying climate and environmental conditions. This enables the evaluation of parameter-
specific impacts, addressing uncertainties associated with soil hydraulic properties and their role in controlling soil moisture
variability. While EOF analysis effectively reveals statistical patterns, its results must be interpreted cautiously, as modes are
mathematical constructs that may not directly correspond to physical processes (Hannachi et al., 2007). The identified patterns
serve as a basis for further analysis, supporting model calibration and validation efforts to enhance the predictive capabilities
of LSMs. To facilitate this analysis, this research employs a Python library for EOF analysis of meteorological, oceanographic,
and climate data developed by Dawson (2016).

**2.3.1   Computation of EOF Using Singular Value Decomposition**

Singular Value Decomposition (SVD) is a robust linear algebra technique widely employed for matrix factorization, enabling
the decomposition of any $n \times m$ matrix, $\mathbf{Y}_w$, without explicitly solving an eigenvalue problem or constructing a covariance
matrix (e.g., Linz and Wang, 2003; Dawson, 2016; Björnsson and Venegas, 1997). In this study, SVD is utilized to compute
the EOF modes by decomposing the matrix of soil moisture anomalies, $\mathbf{Y}_w$, into orthogonal components. The decomposition
is represented as:

$$
\begin{bmatrix} \mathbf{Y}_w : \\ n \times m \end{bmatrix} = \begin{bmatrix} u_{11} & u_{12} & \cdots & u_{1p} \\ u_{21} & u_{22} & \cdots & u_{2p} \\ \vdots & \vdots & \ddots & \vdots \\ u_{n1} & u_{n2} & \cdots & u_{nn} \end{bmatrix} \begin{bmatrix} \gamma_{11} & 0 & \cdots & 0 \\ 0 & \gamma_{22} & \cdots & 0 \\ \vdots & \vdots & \ddots & \vdots \\ 0 & 0 & \cdots & \gamma_{nm} \end{bmatrix} \begin{bmatrix} v_{11} & v_{12} & \cdots & v_{1p} \\ v_{21} & v_{22} & \cdots & v_{2p} \\ \vdots & \vdots & \ddots & \vdots \\ v_{n1} & v_{n2} & \cdots & v_{mm} \end{bmatrix} \tag{1}
$$

$$
\mathbf{Y}_w = \mathbf{U}\mathbf{\Gamma}\mathbf{V}^T, \tag{2}
$$

where $\mathbf{U}$ ($n \times n$) contains the left singular vectors (spatial EOFs), $\mathbf{V}$ ($m \times m$) contains the right singular vectors (temporal
principal components, PCs), and $\mathbf{\Gamma}$ ($n \times m$) is a diagonal matrix with non-negative singular values $\gamma_i$ ($\Gamma_{ij} = \delta_{ij}\gamma_i$). The singular
values $\gamma_i$ quantify the variance captured by each EOF mode, and $\rho = \min(n, m)$ determines the number of non-zero singular
values.



For this analysis, the soil moisture data matrix $\mathbf{Y}_w$ represents the area-weighted, demeaned anomalies simulated by CLM5, with $n$ rows corresponding to temporal steps and $m$ columns representing spatial grid points. To reduce redundancy, we employ

truncated SVD (tSVD), retaining only the top $\rho$ singular values and their associated singular vectors:

$$\mathbf{Y}_w \approx \hat{\mathbf{U}}_\rho \hat{\mathbf{\Gamma}}_\rho \hat{\mathbf{V}}_\rho^T, \tag{3}$$

where $\hat{\mathbf{U}}_\rho$ ($n \times \rho$) contains the leading EOFs, $\hat{\mathbf{\Gamma}}_\rho$ ($\rho \times \rho$) is the diagonal matrix of the largest singular values, and $\hat{\mathbf{V}}_\rho^T$ ($\rho \times m$) represents the corresponding principal components. Singular vectors associated with smaller singular values are discarded, improving computational efficiency while preserving the dominant variability patterns (Figure 3).

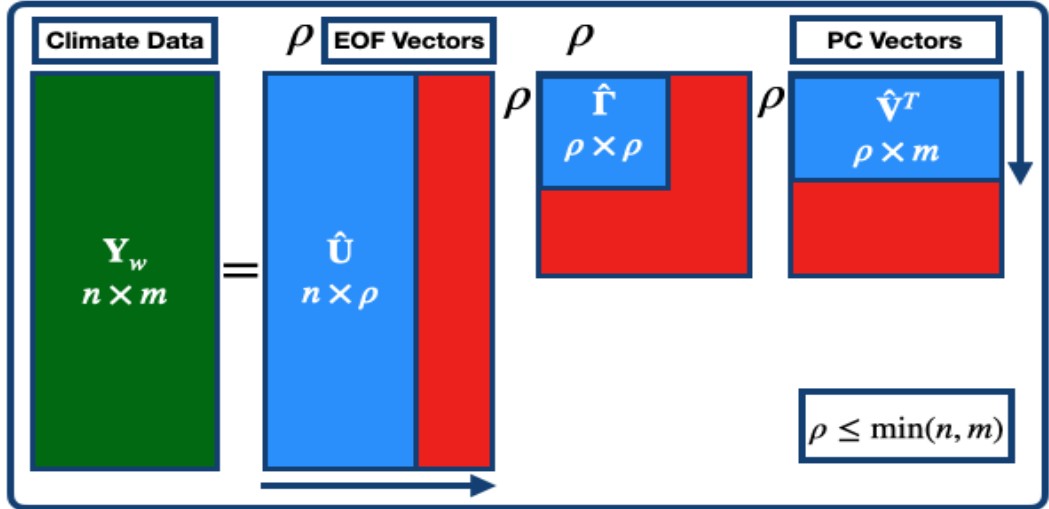

**Figure 3.** tSVD applied to the soil moisture anomaly dataset. The matrix $\mathbf{Y}_w$ ($n \times m$) is decomposed into $\hat{\mathbf{U}}_\rho$ ($n \times \rho$) for EOFs, $\hat{\mathbf{\Gamma}}_\rho$ ($\rho \times \rho$) for singular values, and $\hat{\mathbf{V}}_\rho^T$ ($\rho \times m$) for PCs. The truncation level $\rho$ is chosen such that $\rho \leq \min(n, m)$.

The singular values from tSVD are used to calculate the explained variance ($\%\mathrm{EV}_i$) for each EOF mode, quantifying their contribution to the dataset's variability:

$$\%\mathrm{EV}_i = \frac{\gamma_i}{\displaystyle\sum_{j=1}^{\rho} \gamma_j} \times 100\%, \quad i = 1, 2, \ldots, \rho. \tag{4}$$

The first EOF mode typically explains the largest fraction of variance, representing the dominant spatial pattern, while subsequent modes capture progressively smaller uncorrelated patterns. This hierarchical decomposition provides a powerful frame-

205 work for analyzing spatiotemporal variability in soil moisture anomalies and assessing the relative contributions of soil hydraulic parameters and climate drivers. EOF analysis, through tSVD, ensures that the representation of dominant patterns is efficient and interpretable, enabling robust physical insights into the factors controlling soil moisture variability.



### 2.3.2 Quantifying Similarity of Spatial EOF Modes Using Euclidean Distance

The Euclidean distance metric was employed to assess the similarity or dissimilarity between spatial EOF modes derived from
210 distinct datasets. This metric, commonly used in mathematics and data analysis, calculates the straight-line distance between
two points in Euclidean space, providing a direct and interpretable measure of the geometric proximity between patterns
(e.g., Elmore and Richman, 2001). Its simplicity and intuitive interpretation make it particularly suitable for comparing spatial
variability patterns obtained through EOF analysis. A smaller Euclidean distance indicates a high degree of similarity between
the EOF modes, suggesting a closer alignment of the underlying spatial patterns. Conversely, a larger distance reflects greater
dissimilarity, indicating distinct spatial characteristics or variability between the datasets. In this study, the Euclidean distance
was used to compare the spatial EOF modes from the ERA5-Land reanalysis dataset and the SP-MIP model experiments,
representing different data decomposition results. The Euclidean distance for two spatial EOF modes, $\mathcal{X}$ (ERA5-Land) and $\mathcal{Y}$
(SP-MIP), was computed using the following equation:

$$\mathrm{EucD}(\mathcal{X}, \mathcal{Y}) = \sqrt{\sum_{i=1}^{n} (\mathcal{X}_i - \mathcal{Y}_i)^2}, \tag{5}$$

where $n$ is the number of elements in each spatial EOF mode.

This approach enabled the identification of regions within the CONUS domain where the spatial EOF patterns differed
significantly, highlighting areas requiring improved parameterization of soil properties in land surface models. By quantifying
these differences, the Euclidean distance analysis provides actionable insights into the spatial scales and regions where soil
parameter settings have the most significant impact, thereby supporting targeted model refinements and enhanced soil moisture
simulations.

### 2.3.3 Taylor Diagram for Evaluating Spatial EOF Modes

Taylor Diagrams (TDs) (Taylor, 2001) were applied to assess spatial EOF modes, offering a clear and intuitive visualization
of three essential statistical measures: correlation (COR), standard deviation (STD), and root mean square error (RMSE).
These diagrams are extensively employed in geophysical sciences to evaluate and compare model performance across vari-
230 ous dimensions (e.g., Qiao et al., 2022). Their capability to display the relationship between modeled and observed patterns
simultaneously makes them particularly useful for examining the variability and accuracy of spatial EOF modes derived from
climate datasets. In this research, Taylor diagrams were used to compare the spatial EOF modes of the ERA5-Land reanalysis
dataset against the SP-MIP model experiments. The standard deviation of the ERA5-Land spatial modes served as a bench-
mark for assessing the variability of the SP-MIP modes. The diagrams assessed the similarity of the patterns by using three
metrics: the correlation coefficient, which evaluates the alignment of spatial patterns; the centered RMSE, which measures the
magnitude of pattern differences; and the standard deviation, which indicates the amplitude of variability within each mode.
These combined metrics offer a thorough assessment of spatial pattern differences. Taylor diagrams help identify specific EOF
modes where SP-MIP experiments differ from the ERA5-Land reference, pinpointing areas for possible model enhancement.



By incorporating these metrics into one framework, the diagrams facilitate the focused improvement of soil parameterizations
in land surface models, better capturing essential spatial variability patterns in soil moisture.

## 3 Results and Discussion

### 3.1 Spatial Variability in Annual Mean Soil Moisture Across CONUS

Despite consistent forcing data (GSWP3) and model resolution ($0.5°$), the experiments reveal notable differences in soil mois-
ture spatial patterns due to variations in soil parameter derivation, underscoring the critical role of soil parameters in controlling
simulations. These differences are reflected in the annual mean soil moisture across the CONUS region, which ranged from
$\approx 195 \mathrm{kg\ m}^{-2}$ to $380 \mathrm{kg\ m}^{-2}$, calculated by averaging daily soil moisture from 1980 to 2010 (Figure 4). The spatial distribu-
tion of soil moisture across all experiments reflects well-established precipitation gradients and temperature variability, with
higher soil moisture levels over the central Great Plains and ENA regions and lower values in the arid southwest (WNA).
These findings agree with previous studies documenting the relationship between soil moisture, precipitation, and temperature
in these regions (Welty and Zeng, 2018; Koster et al., 2004; Koukoula et al., 2021; Melillo et al., 2014; Chatterjee et al., 2022).
The pronounced variability in soil moisture in the Great Plains aligns with the principles of continentality, where greater dis-
tances from large water bodies amplify seasonal precipitation and evaporation differences (Gimeno et al., 2010). Among the
experiments, EXP3 (Figure 3d) shows the highest soil moisture levels, followed by EXP2 (Figure 4c) and EXP1 (Figure 4b).
These differences reflect the impact of soil parameter derivation, with EXP1 producing lower soil moisture magnitudes, EXP2
resulting in moderate values, and EXP3 yielding the highest levels.

The results of EXP4 highlight the role of soil texture in modulating soil moisture distribution. For example, EXP4a (loamy
sand, Figure 4e) exhibits low soil moisture in the arid southwest (WNA) and NCA, consistent with the limited water retention
capacity of loamy sand. EXP4b (loam, Figure 4f) shows a more balanced soil moisture distribution, with drier conditions in
WNA and wetter conditions in ENA, reflecting the moderate water holding characteristics of the loam. EXP4c (clay, Figure
4g) shows higher soil moisture levels over ENA due to the high water retention capacity of clay, while EXP4d (silt, Figure 4h)
exhibits heterogeneous soil moisture patterns influenced by environmental variability and the intermediate hydraulic properties
of the silt. These results show that uncertainties in soil parameterization significantly affect soil moisture simulations in the
CLM5 model, consistent with the findings of Brimelow et al. (2010). Our work furthers this research area by systematically
evaluating the role of distinct soil textures (loamy sand, loam, clay, and silt) in shaping soil moisture variability across different
climatic zones. Unlike previous studies, this analysis integrates the spatial distribution of soil moisture with observed climatic
influences, providing a more comprehensive assessment of how parameterization impacts hydrological processes at a continen-
tal scale. Variations in soil parameter settings not only influence soil moisture magnitudes but also alter spatial distributions,
affecting the model's ability to capture hydrological processes at the continental scale. The findings of EXP4 further emphasize
the importance of soil texture in controlling soil moisture distribution, highlighting the need for precise parameterization in
land surface models. This has important implications for improving water resource management, agricultural planning, and
climate impact assessments.





**Figure 4.** Annual mean soil moisture (1980–2010) over the CONUS region, simulated from four experiment types with spatially uniform soil parameter settings: EXP1 (b), EXP2 (c), EXP3 (d), and EXP4 (sub-experiments: EXP4a: loamy sand (e), EXP4b: loam (f), EXP4c: clay (g), and EXP4d: silt (h)). The color bar represents the range of soil moisture values (kg m$^{-2}$), with warmer colors (red and orange) indicating lower soil moisture levels and cooler colors (blue and purple) representing higher soil moisture levels.



## 3.2 Interannual Soil Moisture Anomalies

Interannual root-zone soil moisture anomalies over the CONUS region from 1980 to 2010, derived from CLM5 simulation experiments (EXP1, EXP2, EXP3, and multiple EXP4 configurations) and ERA5-Land reanalysis data, are shown in Figure 5.

Anomalies are computed as deviations from the daily annual mean over the 30-year reference period, following established methodologies for hydrological variability assessment (Tuttle and Salvucci, 2016; Koster et al., 2004; Welty and Zeng, 2018). The top panel of Figure 5 presents anomalies for EXP1, EXP2, EXP3, and ERA5-Land, while the bottom panel includes additional EXP4 parameterizations representing different soil textures (loamy sand, loam, clay, and silt).

Across all configurations, soil moisture anomalies fluctuate around a long-term mean of zero, with values ranging approx-
280 imately from $-20 \text{kg m}^{-2}$ to $+40 \text{kg m}^{-2}$. Positive anomalies signify wetter-than-average conditions, while negative values indicate drier conditions. The CLM5 experiments exhibit pronounced interannual variability, capturing key hydrological extremes, including droughts and wet periods, as observed in ERA5-Land. Notably, ERA5-Land anomalies show higher peaks, particularly in positive extremes, suggesting a possible overestimation of soil moisture in heavily vegetated regions (Lal et al., 2022). Despite these discrepancies, the time-series plots reveal a broad agreement between CLM5 simulations and reanalysis
data in terms of both seasonal and interannual variability.

The relationship between daily soil moisture anomalies from CLM5 and ERA5-Land is further examined in Figure 6. These scatter plots compare CLM5-simulated anomalies with ERA5-Land on a point-by-point basis. The distribution of points is closely aligned along the 1:1 line, with coefficient of determination ($R^2$) values ranging from 0.7 to 0.8 across experiments. These correlations confirm that CLM5 successfully captures the overall variability in ERA5-Land, albeit with some systematic
biases. Specifically, ERA5-Land tends to exhibit larger positive anomalies relative to CLM5, reinforcing the trend observed in the time-series plots. The EXP4 configurations (Figure 6b) show similar performance to EXP1-3, indicating that soil texture variations only moderately impact anomaly correlations at an aggregated scale.

The results indicate significant interannual variability in soil moisture anomalies, with distinct peaks and troughs corresponding to extreme hydrological events. These fluctuations are likely driven by large-scale climatic influences, such as ENSO,
which modulate regional hydrological conditions (Gimeno et al., 2010; Welty and Zeng, 2018). While periodicity in anomalies suggests a possible linkage to climate oscillations, further spectral analysis would be required to confirm such relationships. Additionally, the lack of a discernible long-term trend suggests that soil moisture anomalies remained relatively stable over the study period, with variability largely governed by short to medium-term hydrological cycles. This aligns with findings from Lesinger and Tian (2022), who noted that while interannual fluctuations in soil moisture can be significant, multi-decadal
trends over CONUS tend to be weak or spatially constrained. Overall, the time-series (Figure 5) and scatter plots (Figure 6) collectively demonstrate that CLM5 accurately simulates interannual soil moisture variability in CONUS, with strong correlations to ERA5-Land. However, ERA5-Land's systematic overestimation of positive anomalies highlights a potential bias in reanalysis products, necessitating further evaluation of the mechanisms driving such deviations. Future work should assess regional patterns in soil moisture dynamics and quantify biases across different land cover types to refine model performance.





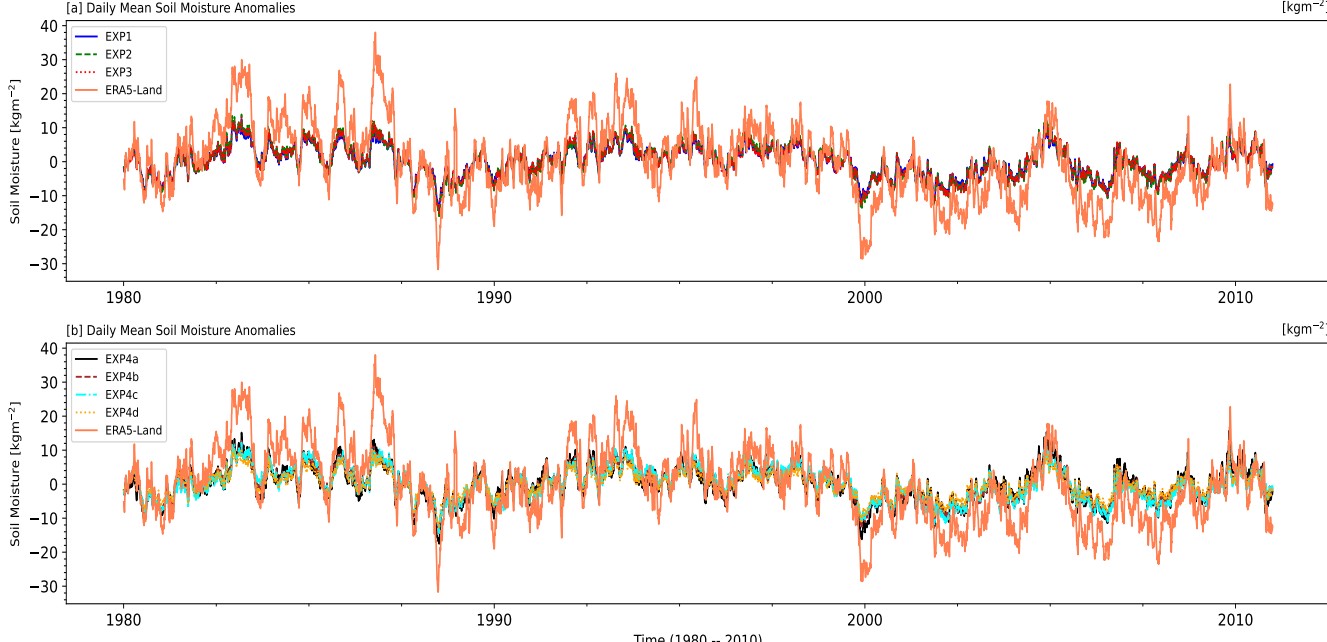

**Figure 5.** Daily soil moisture anomalies from 1980 to 2010 across the CONUS region. (a) Anomalies for EXP1, EXP2, EXP3, and ERA5-Land data. (b) Anomalies for various EXP4 configurations (loamy sand, loam, clay, and silt) alongside ERA5-Land data. Anomalies are calculated as deviations from the daily annual mean over the 30-year period.





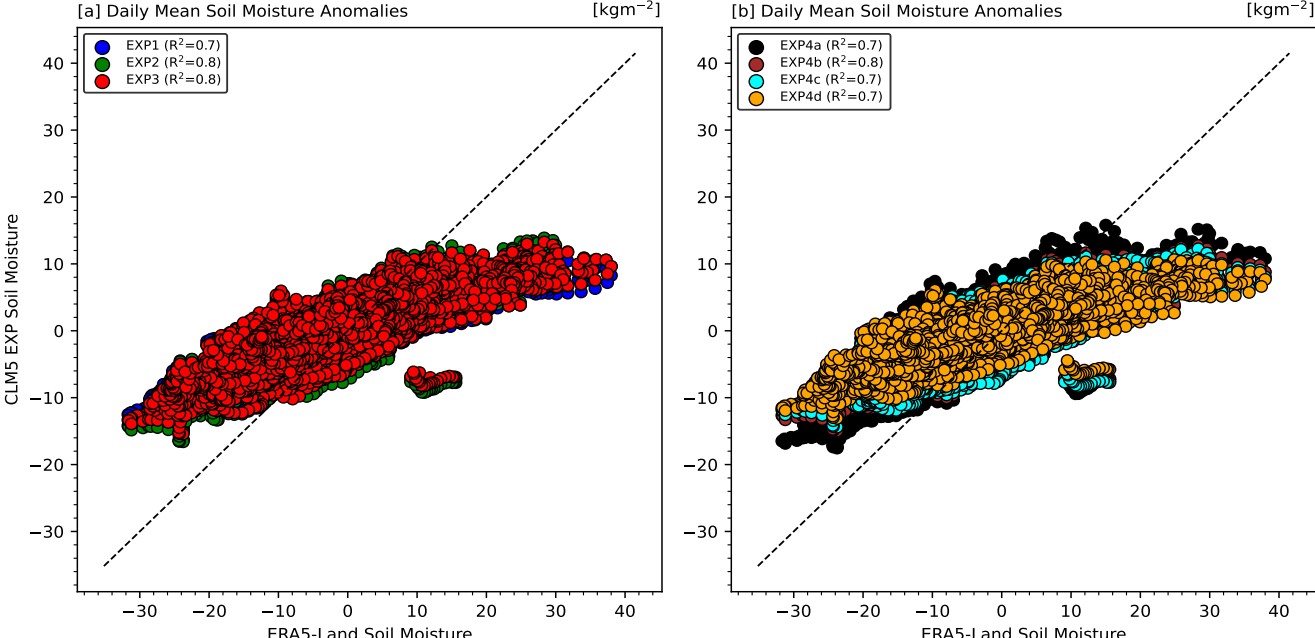

**Figure 6.** Daily mean root-zone soil moisture anomalies for 1980 to 2010 from each CLM5 experiment (EXP1, EXP2, EXP3, and the EXP4 sub-experiments) plotted against ERA5-Land. All anomalies are expressed in [$\text{kg m}^{-2}$]. Each colored marker represents daily anomalies from a given experiment, while the black dashed line denotes the 1:1 relationship. In the legend, $R^2$ values (in parentheses) indicate how closely each experiment's anomalies match those of ERA5-Land.

## 3.3 Seasonal Variability of Soil Moisture

Seasonal cycles of monthly mean soil moisture over the CONUS (1980–2010), comparing CLM5 simulations (EXP1–EXP3 and various EXP4 configurations) with ERA5-Land reanalysis (Figure 7). All models and ERA5-Land exhibit coherent seasonal patterns, characterized by winter maxima and summer minima, aligning with the seasonal interplay of precipitation and evapotranspiration dynamics. These trends are consistent with earlier evaluations of CMIP5 soil moisture simulations (Yuan and Quiring, 2017). Notable inter-experiment discrepancies emerge in the amplitude and phasing of soil moisture variability. For example, EXP4a (loamy sand) displays pronounced springtime fluctuations due to its low water retention capacity, amplifying responses to precipitation and evapotranspiration. Conversely, EXP4c (clay) and EXP4d (silt) exhibit dampened seasonal variability, reflecting their greater water retention efficiency. Systematic biases are evident across experiments: winter and spring soil moisture is underestimated relative to ERA5-Land, while summer values are overestimated. These mismatches likely stem from inaccuracies in soil hydraulic parameterizations governing infiltration, storage, and drainage processes. Interpretive caution is warranted, however, as ERA5-Land itself carries a documented wet bias compared to in situ observations and SMAP satellite retrievals, particularly in densely vegetated areas. This bias also arises from discrepancies in vertical representation: in situ sensors measure moisture at 5cm depth, whereas ERA5-Land integrates the 0–7cm layer, which retains





more moisture post-precipitation Lesinger and Tian (2022). Such differences underscore the need to reconcile observational

and modeled soil moisture definitions. Despite these challenges, the CLM5 simulations robustly reproduce the broad seasonal trends observed in ERA5-Land, affirming their utility for large-scale hydrological analysis. The identified biases, however, highlight critical opportunities to refine soil parameterizations particularly hydraulic properties and vertical layering schemes to better align modeled and observed soil moisture dynamics. Addressing these limitations could enhance the representation of land-atmosphere feedbacks in climate simulations.

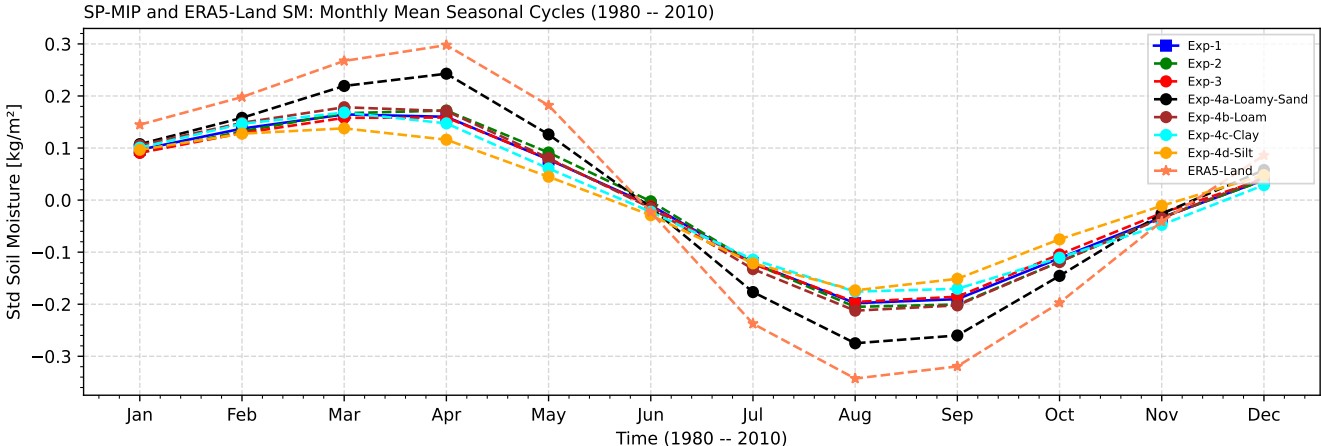

**Figure 7.** Monthly mean seasonal cycles of standardized soil moisture for the period 1980–2010, comparing SP-MIP experiments (EXP1, EXP2, EXP3, and various configurations of EXP4) with ERA5-Land data for the CONUS region. The seasonal cycles exhibit consistent patterns across all experiments and ERA5-Land, with higher soil moisture in winter and lower soil moisture in summer.

## 3.4 EOF Analysis of Soil Moisture Variability

### 3.4.1 Explained Variance and Mode Contributions

This study applies EOF analysis to soil moisture anomalies from the CLM5 simulations (EXP1, EXP2, EXP3) and ERA5-Land data to investigate how soil parameterization influences soil moisture variability in the CONUS region. Figure 6 presents the percentage of variance explained by the first 10 EOF modes for each dataset, illustrating both individual and cumulative

contributions. The EOF modes are ranked by variance percentage, with EOF-1 capturing the highest variance and representing the most significant spatial variability. Across all experiments, EOF-1 explains slightly more variance than EOF-2, suggesting limited separation between these modes and potential mode mixing. The explained variance gradually declines in subsequent modes, with EOF-10 contributing less than 2%, as summarized in Table 3. EOF-1 explains a similar percentage of variance in EXP1 (11.45%) and EXP2 (11.66%), indicating comparable spatial variability patterns. However, in EXP3, EOF-1 captures

only 10.84% of the variance, with mode mixing shifting variance from EOF-1 to EOF-2 (Table 3, arrows). These differences highlight the impact of soil parameterization on representing dominant soil moisture variability. ERA5-Land, serving as a benchmark, exhibits a much stronger EOF-1 contribution (17.5%), emphasizing a more dominant leading mode in observed



data compared to modeled datasets. The cumulative explained variance (Figure 8, green line) further demonstrates the efficiency of the EOF modes in capturing soil moisture variability.

While the first five modes account for about 40% of the variance in ERA5-Land, modeled datasets require approximately six modes to reach the same threshold. This distribution suggests that simulations spread variance more evenly across modes, reflecting differences in spatial patterns between models and observations. To ensure comparability, adjustments aligned the EOF modes across datasets. For instance, shifts in EXP3 and ERA5-Land were necessary to match dominant spatial patterns, such as EOF-1 and EOF-2 swaps (marked in red and blue in Table 3). These adjustments highlight the sensitivity of EOF

rankings to mode mixing and the challenges of directly comparing modeled and observed datasets. In addition, Appendix A (Figure A1) provides additional EOF analysis results for EXP4a-d, detailing variance explained across experiments. The findings reinforce the influence of soil parameterization on the spatial distribution of soil moisture and emphasize the need for improved alignment with observed patterns, as reflected in ERA5-Land.

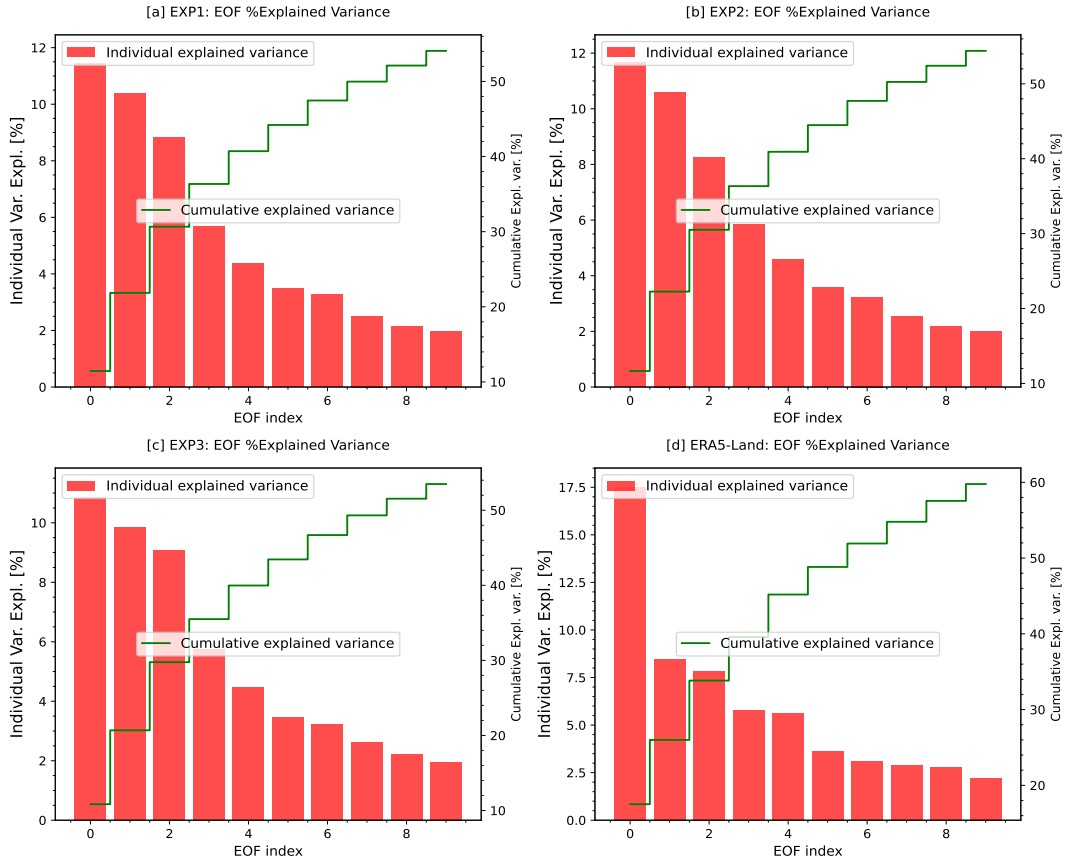

**Figure 8.** The variance explained by each separate and combined EOF in the CLM5 soil moisture experiment is depicted. Red bars represent the contribution of each EOF individually, while the green line shows the cumulative proportion for the initial 10 EOF modes.





**Table 3.** Percentage of variance explained (%Expl. Var.) by the first 10 EOF modes for EXP1, EXP2, and EXP3 model runs, and ERA5-Land benchmark data. Arrows and superscripts indicate EOF mode swaps for consistent comparisons across datasets (see Figure 9).

| EOF Mode | EXP1 %Expl. Var. | EXP2 %Expl. Var. | EXP3 %Expl. Var. | ERA5-Land %Expl. Var. |
|----------|------------------|------------------|------------------|------------------------|
| EOF-1 | 11.45 | 11.66 | 10.84 ↓[2] | 17.5 ↓[2] |
| EOF-2 | 10.40 | 10.60 | 9.85 ↑[1] | 8.48 ↓[3] |
| EOF-3 | 8.81 | 8.25 | 9.08 | 7.83 ↑[1] |
| EOF-4 | 5.69 | 5.83 | 5.73 | 5.75 |
| EOF-5 | 4.37 | 4.59 | 4.48 | 5.61 |
| EOF-6 | 3.49 | 3.56 | 3.48 | 3.64 |
| EOF-7 | 3.26 | 3.23 | 3.24 | 3.10 |
| EOF-8 | 2.51 | 2.53 | 2.63 | 2.86 |
| EOF-9 | 2.14 | 2.16 | 2.22 | 2.76 |
| EOF-10 | 1.96 | 1.99 | 1.95 | 2.22 |

### 3.4.2 Spatial and Temporal Analysis of EOF Modes for Soil Moisture Variability

Spatial distribution of the first three EOF modes from soil moisture anomalies in CLM5 simulations (EXP1, EXP2, EXP3) and ERA5-Land (control). The correlation coefficients, ranging from -1 to +1, indicate the strength and direction of the relationship between EOF patterns and soil moisture anomalies (Figure 9). EOF-1 patterns (Figures 9d, g, j) reveal strong positive correlations in central and southeastern ENA, highlighting a dominant mode of variability. Negative correlations are observed in WNA and CNA, indicating contrasting modes of soil moisture variability in the CONUS region. The variance explained

by EOF-1 ranges from 9.85% (EXP3) to 11.66% (EXP2), with ERA5-Land explaining significantly more variance at 17.5%. These spatial patterns align with large-scale climatic influences such as precipitation gradients and geographic features. For example, Gaffin and Hotz (2000) noted that the Appalachian Mountains exhibit strong precipitation gradients due to storm systems lifting moist southerly winds, enhancing soil moisture in ENA. The corresponding principal components (PC-1; Figure 10a) indicate temporal variability, with notable peaks during 2003 to 2004 and 1988 to 1999, corresponding to documented

climatic events such as ENSO-driven precipitation anomalies (Ye et al., 2023; Gimeno et al., 2010). The close agreement of PC-1 across all experiments highlights the robustness of EOF-1 in representing dominant soil moisture variability, although slight differences suggest sensitivity to parameterizations.

EOF-2 (Figures 9e, h, k) exhibits a distinct dipole pattern, with positive correlations in the central Great Plains and negative correlations over ENA, reflecting a wide spread in soil moisture variability. This dipole nature, which explains 10.40% to

365 10.84% of the variance, is consistent with regional climatic processes such as precipitation and evapotranspiration dynamics influenced by terrain and hydrological conditions. For example, positive correlations in the central Great Plains may result from localized convective precipitation; however, isotope studies indicate that precipitation in this region is influenced by moisture transported from external sources, such as the Gulf of Mexico, rather than solely from local convection (Sanchez-Murillo et al.,



2023). Negative correlations in ENA could reflect the influence of evapotranspiration or soil drainage patterns (Famiglietti,
2014). In particular, EXP3 shows a stronger positive correlation in the desert southwest, indicating a greater sensitivity to
soil parameters in arid regions, which can alter soil water retention and infiltration rates. Furthermore, EOF-3 (Figures 9f, i,
l) highlights localized variability, with positive correlations in the Pacific Northwest and negative correlations over Texas in
CNA. This mode explains less variance than EOF-1 and EOF-2, ranging from 8.25% (EXP2) to 9.85% (EXP3), but captures
important regional processes. The Pacific Northwest patterns may be influenced by orographic precipitation, while negative
correlations in Texas could reflect drought conditions dominated by soil type and fine texture which have a high potential
for water retention (Haverkamp et al., 2005) and fine-texture which have a high potential for water retention. Although the
spatial patterns of EOF-3 are broadly similar between experiments, slight shifts in correlation intensity and location suggest
localized impacts of soil parameterizations. The PCs (Figure 10c) show weaker temporal variability, with occasional peaks
corresponding to distinct climate events, which emphasizes the regional specificity of EOF-3. The appendix includes Figures
A2 and A3, which offer additional results highlighting the spatial and temporal variability of EXP4a-d EOF across experiments,
further supporting the findings discussed. Lastly, the results emphasize the significant role that soil parameterizations play in
soil moisture variability within the CLM5 model. Differences in the spatial and temporal patterns of EOFs indicate the model's
sensitivity to these parameterizations, especially in areas with intricate terrain or significant climate variability. The alignment
of EOF-1 with ERA5-Land underscores the robustness of the model's primary modes, while discrepancies in EOF-2 and
EOF-3 highlight regions where model refinements could enhance localized soil moisture predictions. This study stresses the
importance of improving soil parameterizations to increase the precision of hydrological simulations and effectively capture
the interaction between soil moisture and climatic elements.





**Figure 9.** Spatial patterns of the first three Empirical Orthogonal Functions (EOFs) of soil moisture for the continental United States, derived from ERA5-Land reanalysis data and three different experimental model runs (EXP1, EXP2, EXP3). Panels (a) to (c) represent EOF-1, EOF-2, and EOF-3 from the ERA5-Land dataset, respectively. The color shading indicates the correlation coefficient, with blue representing negative and red representing positive correlations.





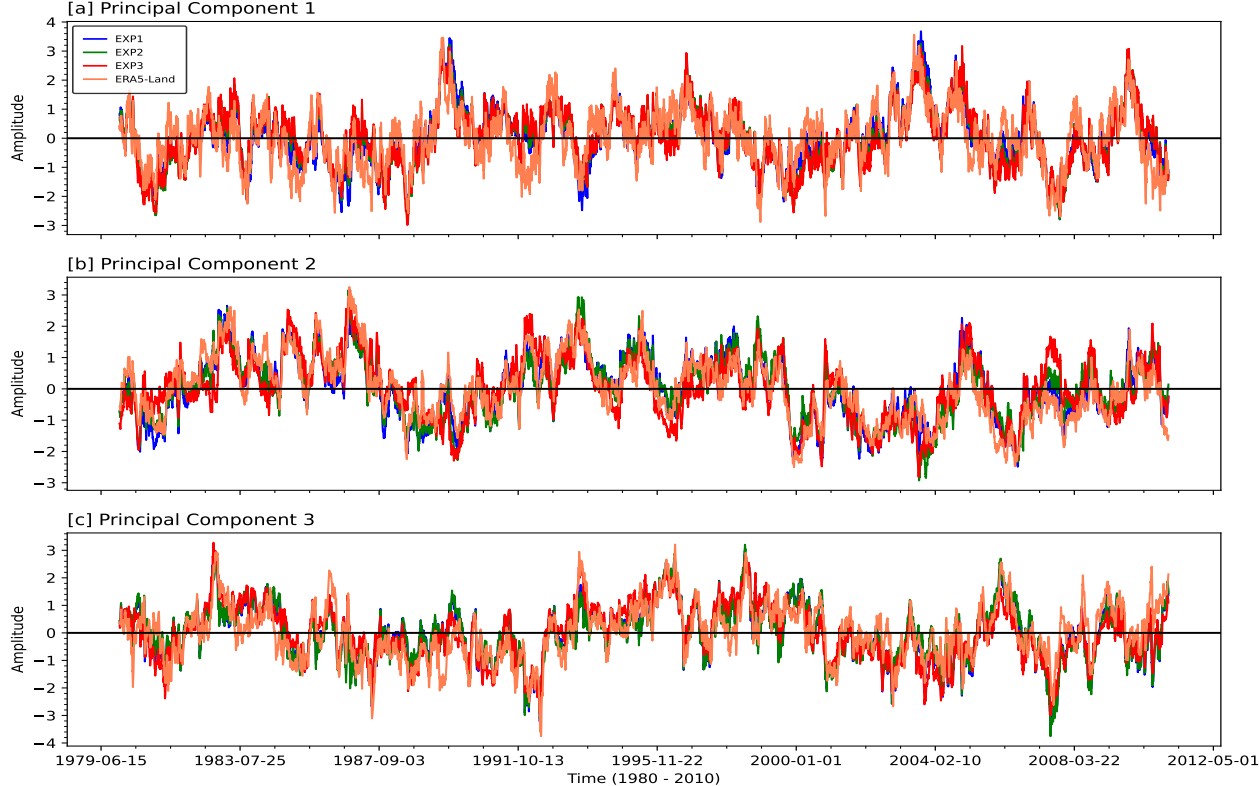

**Figure 10.** Temporal Variability (PC) of corresponding EOF over time (1980-2010) displaying the amplitude of the first four PCs: EXP1 (blue), EXP2 (green), and EXP3 (orange) derived from the soil moisture decomposition respective of their simulation experiments.

### 3.4.3 EOF Modes: Euclidean Distance Analysis

The Euclidean distance between the spatial patterns of EOF modes derived from soil moisture anomalies in CLM5 SP-MIP

model experiments (EXP1, EXP2, and EXP3) and the corresponding EOF modes from the ERA5-Land reanalysis (Figure 11). The Euclidean distance quantifies the dissimilarity between the spatial modes, with smaller values indicating closer agreement with the ERA5-Land benchmark. Regions with hatched lines represent areas where the Euclidean distance falls below a threshold of 5, suggesting a strong alignment between the model-derived EOFs and the observed EOFs in these locations. EOF-1 exhibits the most consistent alignment across experiments, particularly in the western and northwestern portions of the

CONUS region (WNA). The hatched areas in these regions indicate that the spatial variability of soil moisture in these areas is well-represented by the model, reflecting accurate capture of large-scale hydrological processes influenced by precipitation gradients and topographic features (Gaffin and Hotz, 2000; Famiglietti, 2014). In contrast, the central Great Plains consistently shows larger Euclidean distances for all three EOF modes across experiments, suggesting significant discrepancies between the modeled and observed soil moisture patterns in this region. This discrepancy may be attributed to limitations in soil pa-

rameterizations or the complexity of hydrological and climatic processes, such as precipitation variability and soil moisture

 

precipitation feedbacks, as highlighted by Koster et al. (2004) and Welty and Zeng (2018). Compared to ERA5-Land, EXP1 shows a better agreement with ERA5-Land in the WNA region for EOF-1, while the performance in other regions remains mixed across the experiments. EOF-2 and EOF-3 exhibit increased variability in Euclidean distances, with fewer hatched areas, indicating challenges in capturing smaller-scale processes and dipole patterns present in these modes (Hannachi et al., 2007; Monahan et al., 2009). These findings underscore the model's sensitivity to parameterizations and highlight the need for targeted improvements in the central Great Plains and other regions with persistent discrepancies. By refining soil parameter settings and incorporating additional observational constraints, future experiments could achieve better alignment with ERA5-Land, thereby enhancing the accuracy of regional soil moisture simulations (Lawrence et al., 2019; Tuttle and Salvucci, 2016).

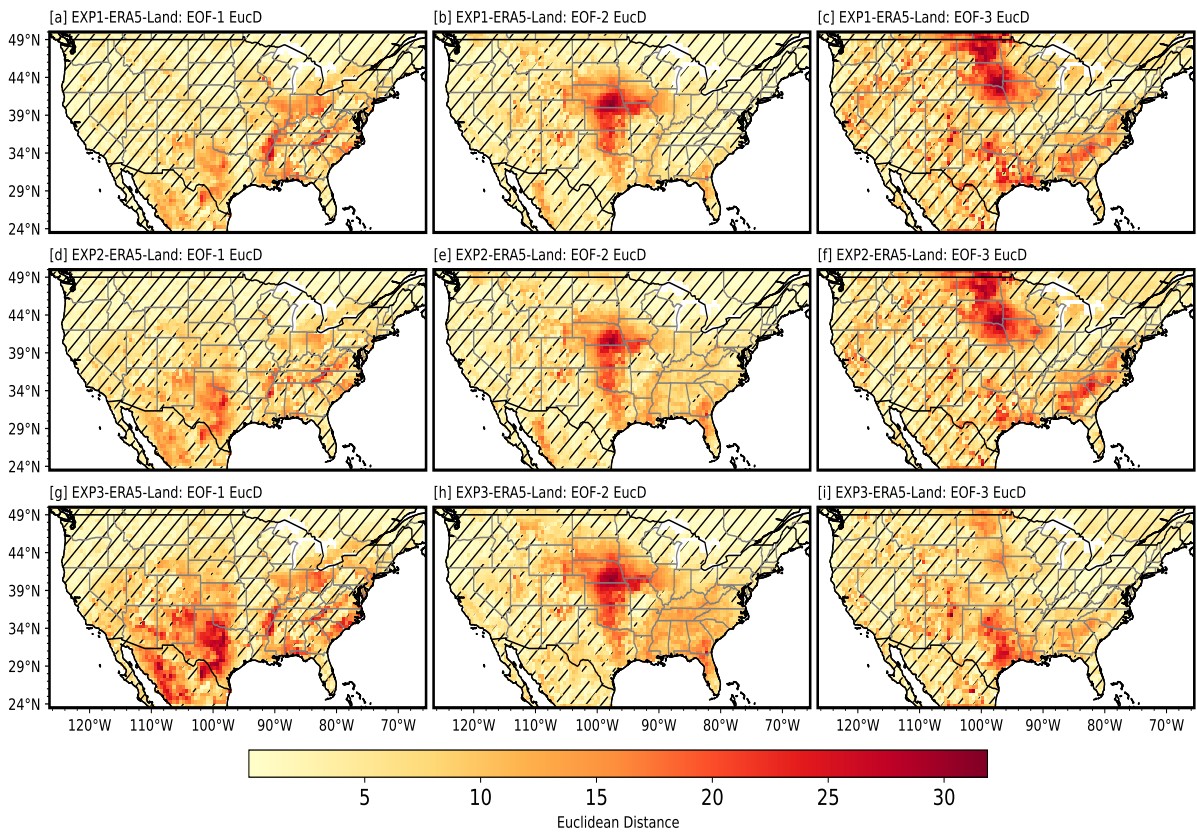

**Figure 11.** Euclidean distance between EOF modes from SP-MIP experiments (EXP1, EXP2, EXP3) and ERA5-Land. Hatched areas indicate regions where the distance is below the threshold of 5, showing closer agreement with ERA5-Land.

### 3.4.4 EOF Modes: Taylor Diagram Analysis

TDs (Figure 12) provide a comprehensive statistical summary of how well EOF patterns from different experiments match those of ERA5-Land by depicting three key statistics: the standard deviation (dotted lines), the correlation coefficient, and the



centered root mean square error (RMSE). Each marker's position on the plot indicates how accurately the soil moisture EOF mode pattern aligns with the ERA5-Land EOF mode. For EOF-1 (Figure 12a), the standard deviations of the EOF modes for all model experiments are relatively close to the reference EOF mode, ranging between 4.0 and 6.5, which suggests a good match in terms of variability. The pattern correlations range between 0.6 and 0.95, with EXP4d demonstrating the highest pattern correlation. This indicates that the spatial pattern of EXP4d aligns more closely with the ERA5-Land EOF mode. In EOF-2 (Figure 12b), the standard deviations remain consistent with the reference EOF mode, while the pattern correlations cluster between 0.4 and 0.7. This highlights a moderate similarity in the spatial patterns of EOF across the experiments and in the reference EOF mode for the second mode of variability. For EOF-3 (Figure 12c), the EOF modes generally exhibit a pattern correlation of around 0.8 and a standard deviation of approximately 5.0. However, the EXP4d EOF deviates, centered around a lower standard deviation of 3.5. These variations emphasize the influence of soil parameter settings in the simulations of the CLM5 model, illustrating how adjustments in these settings affect the alignment of the EOF mode patterns with the ERA5-Land reference EOF mode.

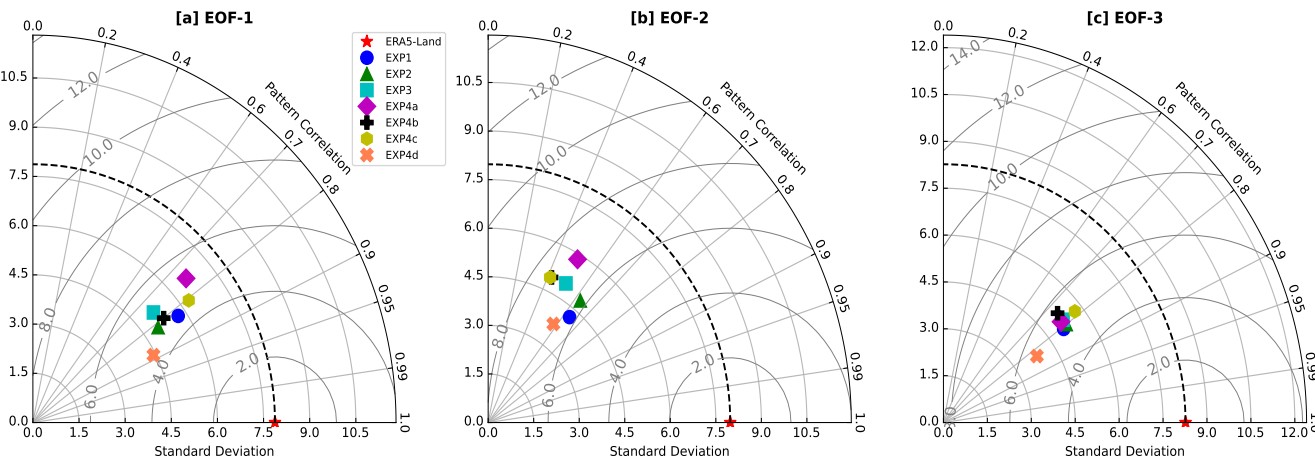

**Figure 12.** Taylor Diagrams (TDs) for the leading three EOFs from multiple experiments (EXP1, EXP2, EXP3, EXP4a, EXP4b, EXP4c, EXP4d) and ERA5-Land. The diagrams summarize standard deviation, correlation coefficient, and RMSE, with marker placement indicating the alignment of modeled EOF modes with ERA5-Land.

## 4 Conclusion and Recommendations

This study investigates the influence of soil parameterizations on soil moisture simulations in the CLM5 across the CONUS for the period 1980 to 2010 using EOF analysis. The analysis compared the CLM5 outputs with the ERA5-Land reanalysis data to identify spatial and temporal variability in soil moisture patterns arising from differences in soil parameter configurations. The results highlighted that EXP3, which used the default CLM5 soil parameters, consistently simulated higher soil moisture levels than other experiments. This finding underscores the model's sensitivity to variations in soil hydraulic properties, such as saturated hydraulic conductivity, soil water retention characteristics, and porosity. Seasonal soil moisture dynamics showed broad



consistency across experiments, peaking in winter due to reduced evapotranspiration, and declining in summer when higher temperatures intensified soil drying. However, distinct differences emerged in the magnitude and phase of seasonal cycles, revealing how variations in soil properties can influence processes such as water retention, drainage, and evapotranspiration fluxes. These insights align with previous research, which demonstrated that soil moisture significantly affects hydrological processes and land-atmosphere interactions, particularly through feedback mechanisms that vary regionally across the United States ((Tuttle and Salvucci, 2016; Koster et al., 2004). Furthermore, the amplified sensitivity observed in the arid and semi-arid regions of the CONUS suggests that these areas may be particularly vulnerable to uncertainties in soil parameterization.

EOF analysis further revealed that the first few modes accounted for the majority of the variance in soil moisture between experiments, and EOF-1 mode decomposed from soil moisture consistently explained the largest proportion. The spatial patterns of the first three EOF modes exhibited similar broad-scale features among the experiments, such as dominant moisture gradients across climatic zones. However, notable differences in explained variance and spatial correlations pointed to the influence of soil parameters on the physical processes driving regional moisture variability. Compared with ERA5-Land data using Euclidean distances and Taylor diagrams, the CLM5 output aligned more closely with observations in WNA, reflecting better model performance in capturing the dynamics of mountainous and arid regions. In contrast, persistent discrepancies in the central Great Plains revealed challenges in representing complex interactions between soil hydraulic properties, precipitation variability, and surface-atmosphere feedbacks. These discrepancies are particularly concerning given the region's susceptibility to extreme hydrological events, including droughts and floods (Koster et al., 2004; Ye et al., 2023). The Great Plains is characterized by a highly variable continental climate, with strong seasonal and interannual fluctuations in precipitation and temperature, leading to frequent shifts between wet and dry extremes (Basara and Christian, 2018; McDonough et al., 2020). This climatic variability makes the region hydrologically complex, requiring accurate representation of soil moisture dynamics for land surface hydrology modeling. Errors in soil moisture estimation can propagate into predictions of crop productivity, water resource availability, and flood risk. The findings suggest that refining soil hydraulic parameterizations, such as incorporating high-resolution soil texture data and accounting for heterogeneity, can significantly improve the predictive capacity of CLM5 and other land surface models for climate studies, ecosystem assessments, and resource management.

To address these challenges and improve the accuracy of soil moisture simulation in CLM5, several strategies are recommended. Refinement of soil moisture variability representation using advanced PTFs or machine learning-based approaches can address uncertainties in soil hydraulic parameters, especially in hydrologically complex regions such as the Great Plains. Expanding the use of high-resolution datasets from satellite missions such as the Soil Moisture Active Passive (SMAP) mission and in situ soil moisture networks will provide robust benchmarks for calibration and validation, reducing biases in model outputs (Famiglietti, 2014). Conducting region-specific calibration of soil parameters and comparative multi-model analyses will help address inter-model variability and optimize simulations for diverse climatic zones. Linking soil moisture variability to dynamic vegetation feedbacks can improve the representation of evapotranspiration processes, as vegetation significantly influences soil moisture and water exchange dynamics (Oleson et al., 2010; Ye et al., 2023). Establishing stronger connections between soil moisture variability and large-scale climatic drivers such as the ENSO can enhance seasonal forecasts and long-



term predictive capabilities (Gimeno et al., 2010; Tuttle and Salvucci, 2016). Understanding these links will facilitate better integration of climatic variability into land surface modeling frameworks.

Importantly, these findings also open the door to future efforts that incorporate dynamic soil properties into LSMs. Much of this work demonstrates the dynamism of soil properties, and while this study advances modeling by revealing the impor-
tance of their inclusion, the next crucial step will be developing approaches that allow these properties to be dynamic within LSMs. This paper serves as a foundational step toward that goal, paving the way for more complex and integrated modeling frameworks that better capture soil-hydrology-climate interactions. These recommendations aim to address existing challenges in soil moisture modeling and improve the predictive capabilities of land surface models such as CLM5. Advancing soil hydraulic parameterization and leveraging state-of-the-art observational datasets will enable models to more accurately capture
large-scale hydrological dynamics and localized soil-climate interactions. This, in turn, will support improved water resource management, agricultural planning, and climate adaptation strategies, ultimately contributing to the larger goals of sustainable development and climate resilience.

*Code and data availability.*  All datasets used in this study are publicly accessible through the CLM5 Model SP-MIP simulation data are available for download at ftp://sp-mip:sp-mip2017@data.iac.ethz.ch (Gundmundsson and Cuntz, 2017), along with experimental design de-
tails. This includes files on soil parameters and soil texture for EXP1, EXP2, and EXP4a–d. Additionally, the ERA5-Land can be freely accessed at https://doi.org/10.24381/cds.e9c9c792 (Muñoz-Sabater et al., 2021). The code used to process the data, perform the EOF analyses, and generate the results is available on Zenodo at https://doi.org/10.5281/zenodo.14888812 (Silwimba, 2025). The Zenodo repository provides comprehensive documentation and instructions for reproducing the analysis, and any future updates or additional scripts will be hosted there. For any difficulties in accessing these data or code, or for requests for further information, please contact the corresponding
author.



## Appendix A

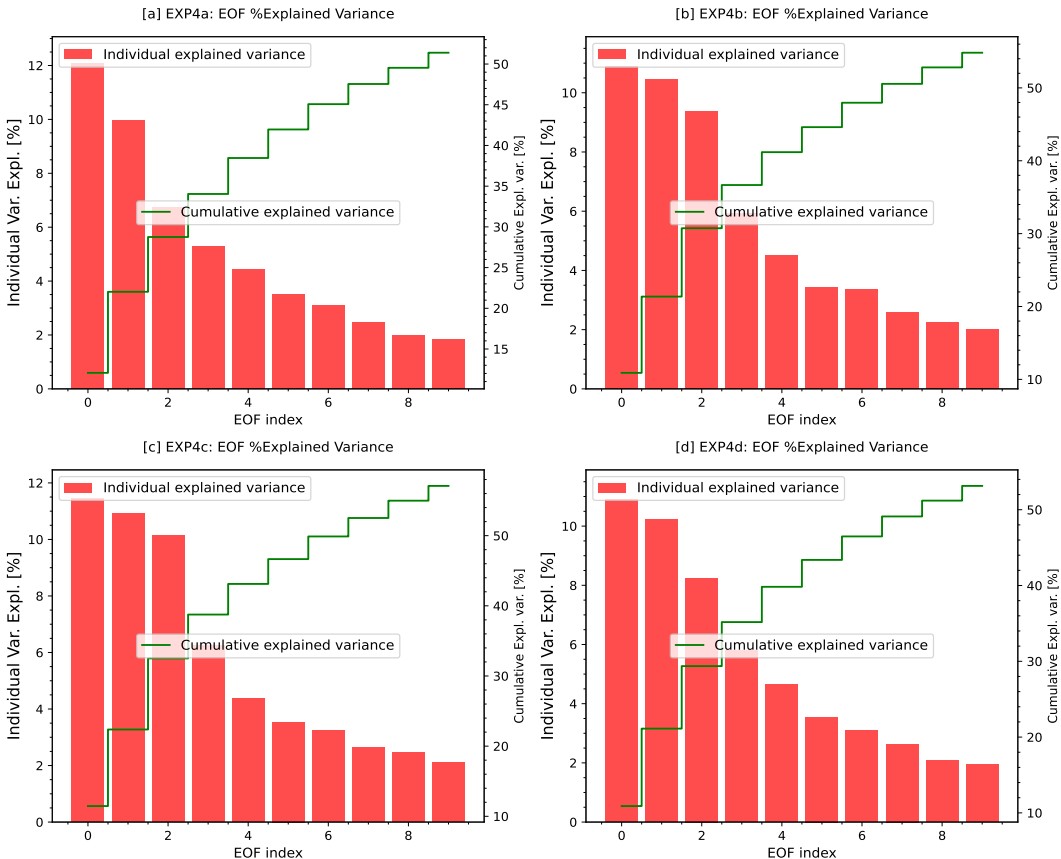

**Figure A1.** Contribution of Variance by Individual and Cumulative EOFs in CLM5 Soil Moisture Experiments. The red bars indicate the portion of variance each separate EOF mode accounts for, whereas the green line depicts the cumulative percentage of variance explained by the first ten EOF modes. These plots reveal the significant impact of the early EOF modes in accounting for variance. Panels (a) to (d) relate to different experimental configurations or scenarios, offering a comparative assessment of EOF variance contributions.



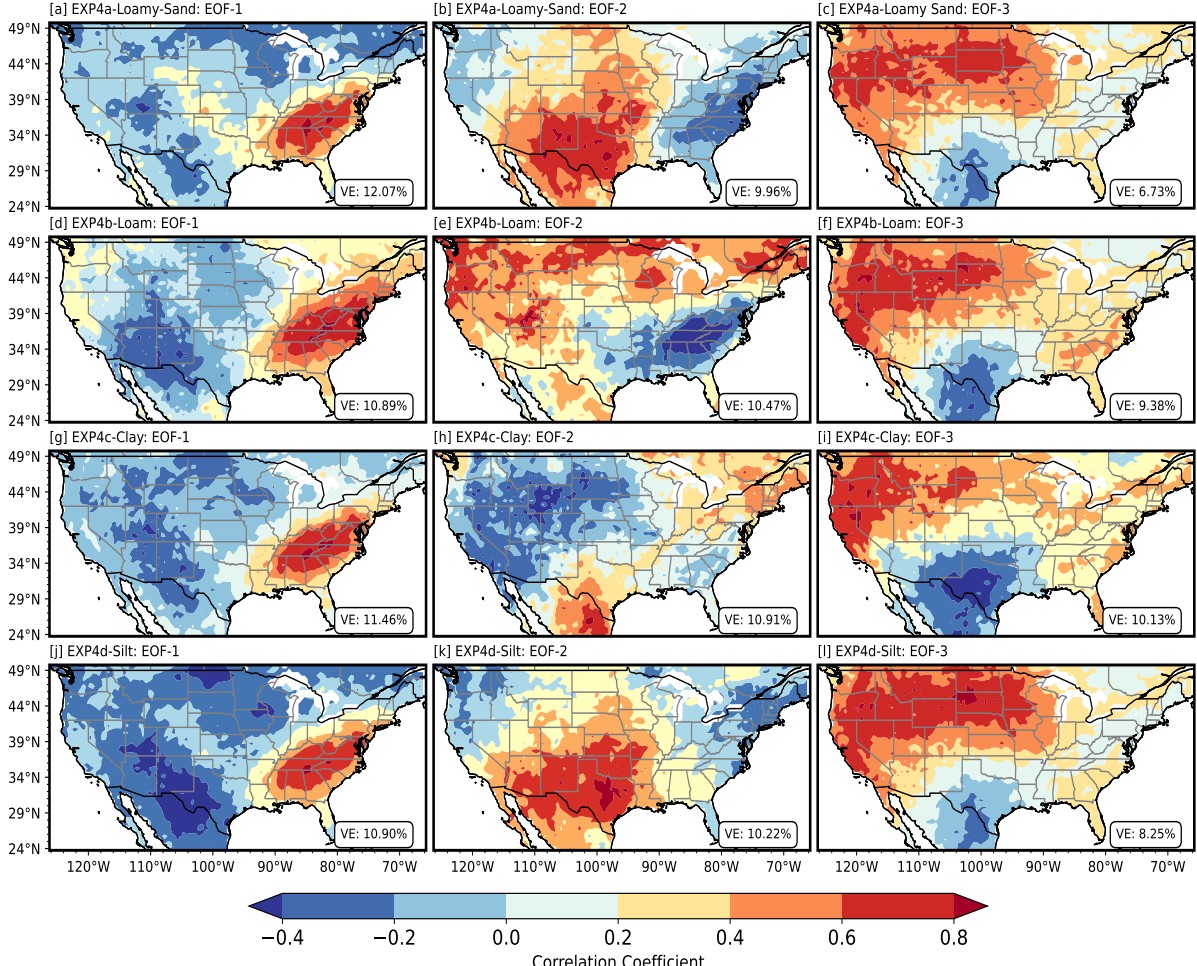

**Figure A2.** The initial three EOF modes' spatial correlation patterns resulting from soil moisture decomposition across the CONUS region are depicted. Panels (a, b, c) are for Experiment 4a (Loamy Sand), (d, e, f) for Experiment 4b (Loam), (g, h, i) for Experiment 4c (Clay), and (j, k, l) for Experiment 4d (Silt). Each panel displays the first (EOF-1), second (EOF-2), and third (EOF-3) EOF modes, respectively. Within each panel, the variance explained (VE) by the EOF modes is noted, with EOF-1 consistently exhibiting the highest VE across all soil types. The color gradient (blue to red) indicates the correlation coefficient, with red signifying a strong positive correlation and blue a strong negative one. These patterns showcase how soil moisture decomposition's spatial variability is shaped by distinct soil textures.





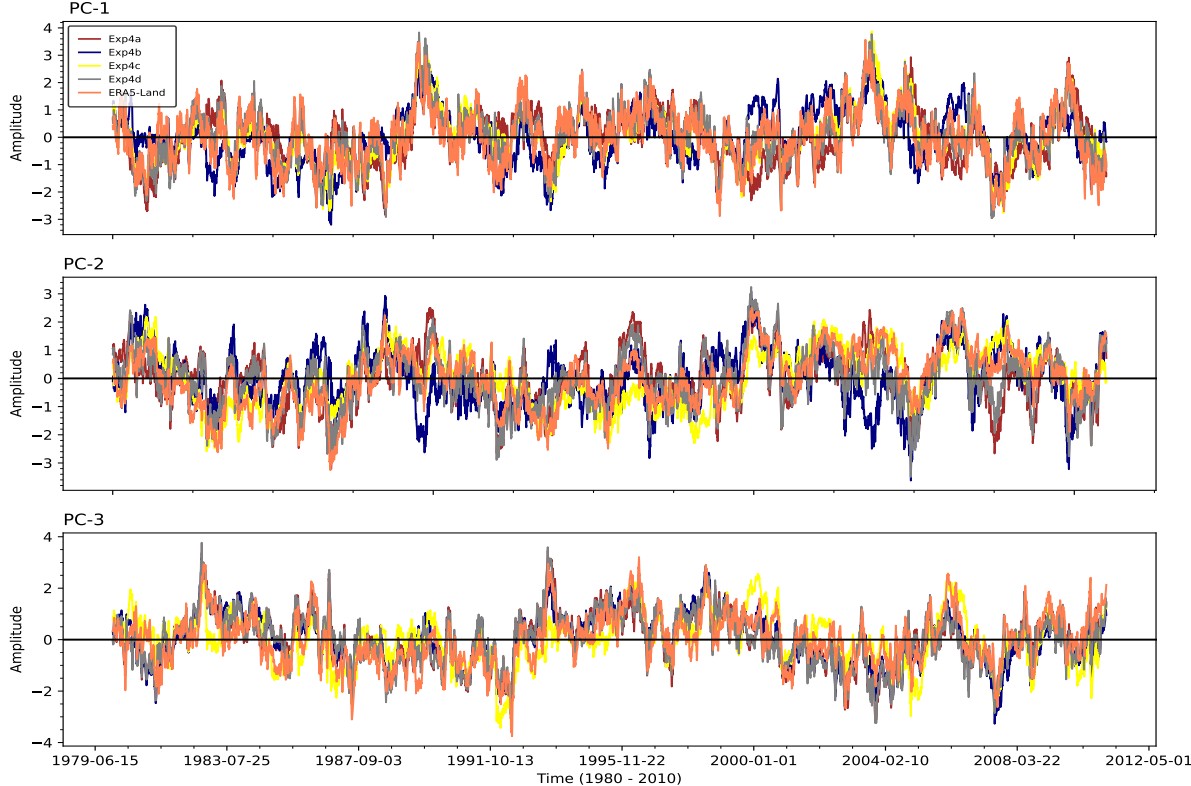

**Figure A3.** Temporal variability of principal components (PCs) derived from the EOF analysis. The plots display the amplitude of the first three principal components: PC-1, PC-2, and PC-3. Each line corresponds to one of the four experimental setups (EXP4a, EXP4b, EXP4c, and EXP4d) or the ERA5-Land reanalysis. PC-1 (top panel) captures the dominant mode of variability, while PC-2 (middle panel) and PC-3 (bottom panel) represent the secondary and tertiary modes, respectively. The x-axis shows the time period (1979–2012), and the y-axis indicates the standardized amplitude. These plots highlight the temporal dynamics of soil moisture variability as captured by different experimental configurations, providing insights into their agreement and divergence relative to the ERA5-Land reference data.





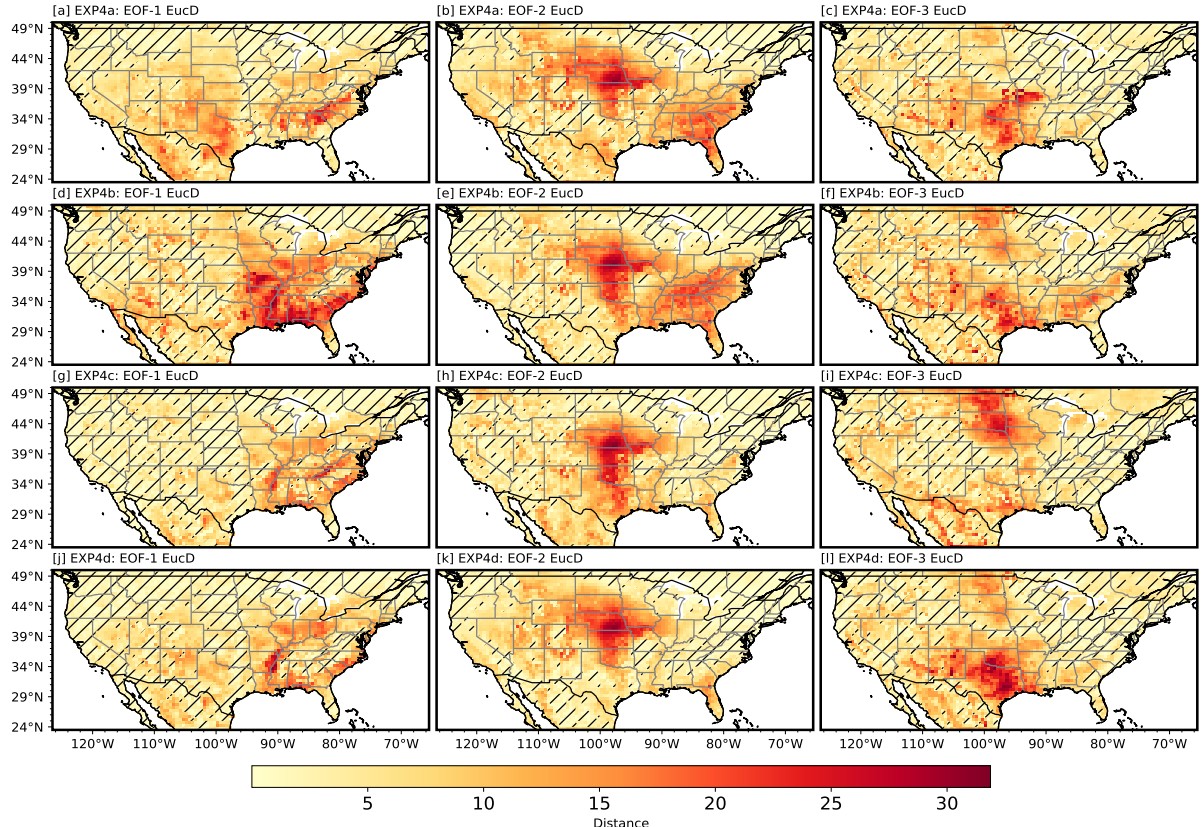

**Figure A4.** The Euclidean distance between EOF modes from SP-MIP experiments (EXP4a, EXP4b, EXP4c, EXP4d) and ERA5-Land is depicted. Panels (a–c) illustrate results for Experiment 4a (Loamy Sand), while panels (d–f), (g–i), and (j–l) pertain to Experiments 4b (Loam), 4c (Clay), and 4d (Silt), respectively. Each column showcases one of the first three EOF modes: EOF-1, EOF-2, and EOF-3. The color bar represents the Euclidean distance, where lower values (yellow) reflect stronger alignment with ERA5-Land, whereas higher values (red) denote more significant discrepancies. Regions with hatched patterns signify distances less than 5, emphasizing areas where the experiments closely align with the ERA5-Land data. These observations reveal the spatial variability in model performance across different soil hydraulic parameter settings and EOF modes.

*Competing interests.* The authors declare that they have no conflicts of interest.

*Acknowledgements.* We acknowledge funding support from the National Science Foundation Frontiers Research in Earth Science program (HA & DRH - 2121760; PLS - 2121694; SAB - 2121639; LL - 2121621; AF - 2121595). Additional support was provided by the Signals
in the Soil (SitS) grant from the USDA National Institute of Food and Agriculture (HA & DRH - no. 2021-67019-34341; SAB - no. 2021-



67019-34338; ANF - no. 2021-67019-34340) and by the National Science Foundation under Grant Nos. 2034232 (PLS) and 2034214 (LL). We also acknowledge support from the National Science Foundation EAR program under Grant Nos. 2034232 and 2121694.

We thank Jesse Nippert for valuable feedback, which helped improve this work.

Any opinions, findings, and conclusions or recommendations expressed in this material are those of the authors and do not necessarily reflect the views of the National Science Foundation.



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
