# Peer review of "Soil Parameterization in Land Surface Models Drives Large Discrepancies in Soil Moisture Predictions Across Hydrologically Complex Regions of the Contiguous United States"

_EGUsphere, 2025_

## Referee Comment (RC2)

**Reviewer Comments for: "Soil Parameterization in Land Surface Models Drives Large Discrepancies in Soil Moisture Predictions Across Hydrologically Complex Regions of the Contiguous United States" by Silwimba et al.**

The authors present a comprehensive and methodologically rigorous study examining the influence of soil hydraulic and textural parameters on soil moisture simulations in CLM5, using soil parameter sets from the Soil Parameter Intercomparison Project (SP-MIP). Model outputs are compared against the ERA5-Land dataset as a benchmark. The study utilizes various analytical approaches—including means and variability assessments (Figures 4 and 5), as well as Empirical Orthogonal Function (EOF) analysis—to investigate dominant spatial patterns and variability in soil moisture across the CONUS region. The findings suggest that soil parameterization has a substantial impact on CLM5 simulations, with notable discrepancies from ERA5-Land, particularly in hydrologically complex regions such as the Great Plains. The default CLM5 setup captures mean climatological patterns reasonably well but tends to underestimate interannual and seasonal variability.

Overall, the manuscript is well-structured, and the English is of generally high quality. The authors have executed a wide array of experiments that meaningfully contribute to our understanding of soil parameter sensitivities in land surface modeling. However, several critical issues remain that merit further investigation before the manuscript is suitable for publication. I recommend **major revisions** to address the following points:

1) **Limited Benchmarking Against Reference Data**

   The exclusive use of ERA5-Land as a benchmark is insufficient. While the authors acknowledge some of ERA5-Land's limitations, it remains a reanalysis product with inherent model dependencies and does not assimilate in-situ soil moisture observations directly. Prior studies [*Koster et al.*, 2009] have demonstrated that soil moisture estimates are highly model-dependent. The validity of conclusions based solely on a single reference dataset is therefore limited.

   To strengthen the analysis, I strongly recommend incorporating additional observation-based datasets, such as GLEM v3 [*Martens et al.*, 2017], SMERGE [*Tobin et al.*, 2019], and MERRA 2 [*Reichle et al.*, 2017]. Each offers distinct advantages—GLEAM and SMERGE incorporate satellite-based observations, whereas MERRA-2 is a reanalysis-based soil moisture data. A recent study [*Duan et al.*, 2025] has shown ERA5-Land's underperformance compared to these alternatives for sub-seasonal to seasonal forecast validations.

2) **Underestimation of Interannual and Seasonal Variability**

   Figures 5 and 7 clearly indicate that all CLM5 configurations substantially underestimate soil moisture variability relative to ERA5-Land. Notably, Figure 5 reveals a tight clustering of CLM5 experiments, suggesting low variability in contrast to the wider spread of ERA5-Land data. However, this important point is underexplored in the manuscript. For instance, lines 284–285 state: "Despite these discrepancies, … broad agreement," which downplays the observed discrepancies.

This variability gap warrants a deeper investigation and further supports the need for multiple observational references (as per Comment 1).

**(3) Neglect of Irrigation Effects**

The authors identify significant differences between CLM5 and ERA5-Land EOF modes in agriculturally intensive areas, particularly the central U.S. (Figure 11b). These "hotspots" overlap spatially with known heavily irrigated regions, including the Ogallala Aquifer and Mississippi Valley [*McDermid et al.*, 2023].

However, the manuscript does not clarify:

- Whether Irrigation was included in CLM5 simulations.

- How Irrigation may affect soil moisture in ERA5-Land.

This omission weakens the attribution of model-observation discrepancies solely to soil parameterization. Explicit discussion on irrigation modeling and its inclusion or exclusion is essential to substantiate the attribution claims made (e.g., Lines 14–15).

**(4) Initial Conditions Not Explained**

The setup of initial conditions in the model simulations remains unclear. Since each experiment involves different soil parameter settings, it is essential that the model reaches equilibrium separately for each case [*Kennedy et al.*, 2024]. Without spin-up or appropriate initialization, differences in the initial soil moisture state could propagate and bias the results.

Please clarify whether each experiment was initialized to equilibrium independently, and if so, provide methodological details.

**Minor Comments:**

- **Line 32**: The phrase "such as artificial neural networks" requires a reference. A citation demonstrating the use of ANN as PTFs would be appropriate.

- **Line 77**: Parentheses are inconsistent in references; e.g., fix "Ji et al……, Zeng et al., (2021)" to consistent formatting.

- **Line 132**: Typo: "PFTS" should be corrected to "PTFs."

- **Lines 160–179 (Section 2.3)**: Consider revising for clarity. There are some repetitions, e.g., "dominant variability modes and their temporal patterns"…" spatial and temporal patterns"…

- **Line 193**: The term "demeaned" could be clarified or rephrased for general readability.

- **Figure 5**: Further explanation is needed regarding the variability difference between ERA5-Land and CLM5 simulations.

- **Lines 218–319**: The statement about 5 cm in-situ sensors vs. ERA5-Land's 0–7 cm integration is not directly relevant when the analysis uses 0–1 m averaged soil moisture. Clarify or remove.

- **Figure 7**: Please explain the significant differences between EAR5-Land variability and CLM5 cluster, except for Exp4-a.

- **Figures 9 and A2**: Clarify whether maps show EOF loadings or correlation coefficients. If correlation coefficients are used, explain the meaning and implications, e.g., correlation with respect to what?

- **Table 3**: Please include a total cumulative variance explained by the first 10 EOF modes for each experiment.

References

Duan, Y., S. Kumar, M. Maruf, T. M. Kavoo, I. Rangwala, J. H. Richter, A. A. Glanville, T. King, M. Esit, and B. Raczka (2025), Enhancing sub-seasonal soil moisture forecasts through land initialization, npj Climate and Atmospheric Science, 8(1), 100.

Kennedy, D., K. Dagon, D. M. Lawrence, R. A. Fisher, B. M. Sanderson, N. Collier, F. M. Hoffman, C. D. Koven, S. Levis, and K. W. Oleson (2024), One-at-a-time parameter perturbation ensemble of the community land model, version 5.1, Authorea Preprints.

Koster, R. D., Z. Guo, R. Yang, P. A. Dirmeyer, K. Mitchell, and M. J. Puma (2009), On the nature of soil moisture in land surface models, J Climate, 22(16), 4322-4335.

Martens, B., D. G. Miralles, H. Lievens, R. van der Schalie, R. A. M. de Jeu, D. Fernandez-Prieto, H. E. Beck, W. A. Dorigo, and N. E. C. Verhoest (2017), GLEAM v3: satellite-based land evaporation and root-zone soil moisture, Geosci Model Dev, 10(5), 1903-1925, doi: 10.5194/gmd-10-1903-2017.

McDermid, S., M. Nocco, P. Lawston-Parker, J. Keune, Y. Pokhrel, M. Jain, J. Jägermeyr, L. Brocca, C. Massari, and A. D. Jones (2023), Irrigation in the Earth system, Nature Reviews Earth & Environment, 1-19.

Reichle, R. H., C. S. Draper, Q. Liu, M. Girotto, S. P. P. Mahanama, R. D. Koster, and G. J. M. De Lannoy (2017), Assessment of MERRA-2 Land Surface Hydrology Estimates, J Climate, 30(8), 2937-2960, doi: 10.1175/Jcli-D-16-0720.1.

Tobin, K. J., W. T. Crow, J. Dong, and M. E. Bennett (2019), Validation of a new root-zone soil moisture product: Soil MERGE, IEEE Journal of Selected Topics in Applied Earth Observations and Remote Sensing, 12(9), 3351-3365.

---

## Author Comment (AC4)

**RESPONSE TO REVIEWER #1 FOR GEOSCIENTIFIC MODEL DEVELOPMENT:**
**MANUSCRIPT EGUSPHERE-2025-713**

We sincerely thank Reviewer #1 for their thorough and constructive feedback on our manuscript, "Soil Parameterization in Land Surface Models Drives Large Discrepancies in Soil Moisture Predictions Across Hydrologically Complex Regions of the Contiguous United States." The comments have significantly helped us identify areas for improvement, and we outline below how we will address each point in the revised manuscript to enhance clarity, robustness, and alignment with the standards of the hydrology and land surface modeling communities. Our responses to the suggestions are detailed below (in blue).

Silwimba et al. investigate the impact of different soil hydraulic parameter sets derived from various approaches on soil moisture variability across the contiguous United States (CONUS), using the Community Land Model version 5 (CLM5). The study employs Empirical Orthogonal Function (EOF) analysis to extract dominant spatiotemporal patterns in soil moisture and assesses how variability in soil hydraulic parameters influences hydrological processes.

The manuscript is generally well-written, with clearly articulated objectives and a methodologically sound design. The topic is timely and of interest to the hydrology and land surface modeling communities. The authors have presented the results in a clear and coherent manner. However, I have a few concerns and suggestions that, if addressed, could improve the clarity and robustness of the manuscript.

**Major and Minor Comments**

1. **Experimental Design Clarity:**

The description of the experimental setup, particularly EXP1, EXP3, and EXP4a–4d, requires further clarification:

   a) For **EXP1**, how exactly does the use of uniform soil hydraulic parameters demonstrate a reduction in inter-model variability? A more detailed explanation of the hypothesis and expected behavior would be helpful.

   **Response:** We appreciate the reviewer's observation and agree that the original wording was potentially misleading. In the revised manuscript, we clarified that EXP1 is not designed to assess inter-model variability per se, since our study uses only CLM5. Instead, EXP1 serves as a baseline control simulation within CLM5, applying globally standardized soil hydraulic parameters derived from SP-MIP uniformly across the CONUS domain. By eliminating spatial variability in soil properties, this setup allows us to isolate CLM5's intrinsic response to a consistent parameter set. The objective is to establish a stable reference point against which the

effects of varying parameterizations in other experiments (EXP2–EXP4) can be compared. We have removed references to inter-model variability and now explicitly define EXP1's role in highlighting intra-model sensitivity to soil parameter changes.

b) For **EXP3**, it is not entirely clear how this experiment isolates the intrinsic inter-model variability. Please elaborate.

**Response:** Thank you for highlighting this confusion. We have revised the description of EXP3 to clarify that it is not meant to isolate "inter-model" variability, but rather to assess CLM5's default behavior using its native parameter configuration. EXP3 uses CLM5's built-in soil maps and lookup tables to assign hydraulic properties, reflecting the model's operational configuration without externally imposed constraints. The goal is to establish a benchmark for CLM5's default performance, allowing us to compare its outputs with more controlled or hypothetical scenarios (e.g., EXP1 and EXP4a-4d). We have removed any mention of "intrinsic inter-model variability" and now focus on evaluating how CLM5's default parameter assumptions influence soil moisture outputs.

c) For **EXP4a–4d**, I find it difficult to understand how four different soil categories are implemented in the model. Do you run the model separately for each soil category? Are these scenarios simulated using four distinct parameter sets applied uniformly across the domain, or are they spatially varying? Clarifying how these simulations were configured in CLM5 is essential.

**Response:** We agree that the initial description of EXP4a–4d lacked sufficient detail. In the revised manuscript, we now clearly state that each of the four experiments (EXP4a–4d) involves a separate CLM5 simulation, in which a uniform set of soil hydraulic parameters corresponding to a specific USDA soil class (loamy sand, loam, clay, silt) is applied consistently across all grid cells in the CONUS. These parameter sets are sourced from SP-MIP and derived using standard PTFs. There is no spatial variation in soil properties within each experiment; the only difference across experiments is the texture class used. This design enables a clean comparison of how different soil textures influence soil moisture and energy balance outputs under identical meteorological conditions. We have revised the text to reflect this modeling setup and its role in evaluating texture-specific sensitivity within CLM5.

1. **Model–Observation Comparison**:
Have the authors considered validating the model outputs against observational soil moisture datasets? Including such comparisons would strengthen the findings and contextualize model performance.

**Response:** We thank the reviewer for this thoughtful suggestion. In the current study, we compare CLM5 outputs to the ERA5-Land reanalysis product, which assimilates a wide range of observational and model-derived inputs to produce a spatially and temporally consistent estimate of soil moisture across CONUS. ERA5-Land was selected for its comprehensive coverage, high temporal resolution, and comparability with the spatial resolution of our CLM5 simulations. The

purpose of this comparison is to examine similarities and differences between the modes of soil moisture variability from the SP-MIP numerical experiments and those of a reanalysis dataset. To carry out this comparison, we use multiple statistical metrics, including Euclidean distance, Taylor Diagrams, and EOF-based analyses (Sections 3.1–3.4), to systematically assess the agreement between CLM5 simulations and ERA5-Land data.

We fully agree, however, that incorporating direct observational datasets such as in-situ measurements from soil moisture monitoring networks or remote sensing products like the Soil Moisture Active Passive (SMAP) mission would provide an additional layer of comparison and help benchmark model performance more rigorously. We underscore that the primary purpose of this comparison is not so much a validation of the model performance, but rather an assessment of the degree to which the ranges of variability in SP-MIP experiment soil moisture response compares with those seen in observational datasets. We admit that some of the comparison metrics we used are commonly used in more strict model validation studies, where benchmarking against multiple datasets is important to evaluate the veracity of model predictions. However, we believe that introducing additional comparisons with observational data may unnecessarily expand the scope of the current work, which seeks primarily to assess the sensitivity of the model to soil parameters.

However, in acknowledgment that the previous version of the manuscript may not have been clear about the purpose of this work, we have added a paragraph in the revised Section 4 acknowledging the narrow scope of our work and emphasizing the value of integrating direct observational datasets in future works, which might seek to more deliberately calibrate soil parameters within a global land model. We note that upcoming efforts will focus on leveraging SMAP and in-situ data to complement reanalysis-based validation and further improve the robustness and interpretability of soil moisture simulations in CLM5. This will support more comprehensive model evaluation and enhance confidence in land surface model applications.

2. **Figure Reference – Line 328:**
   The text refers to Figure 6, but the description seems to match the content of Figure 8. Please verify and correct this reference.

   We thank the reviewer for pointing out the incorrect figure reference on line 328. The text description corresponds to Figure 8, not Figure 6. We have corrected the reference in the revised manuscript to ensure consistency between the narrative and the associated figure.

3. **Regional Subdivisions of CONUS:**
   While the manuscript defines subregions within CONUS, the analysis appears to be conducted solely at the national scale. What is the purpose of introducing these subdivisions if no region-specific results are discussed?

   **Response:** To adrress the reviewer's concern regarding the relevance of regional subdivisions, we have revised Section 2.1 to explicitly clarify the role of the CONUS subregions Western North America (WNA), Central North America (CNA), Eastern North America (ENA), and North Central America (NCA) in our analysis. These subdivisions, based on Giorgi and Francisco (2000), now

serve as a physically meaningful framework for interpreting region-specific patterns in soil moisture variability identified through EOF analysis. The revised text emphasizes how each region captures distinct hydroclimatic characteristics, supporting a spatially disaggregated evaluation of model sensitivity to soil hydraulic parameters. By linking EOF-derived spatial modes and observed model-observation discrepancies (e.g., higher soil moisture in ENA and lower model agreement in CNA) to these macro-regions, the updated section enhances both the interpretability and process-level insights of the results. This regionalization strengthens the methodological coherence of the study and directly supports our objective of understanding spatially heterogeneous parameter impacts across the CONUS domain.

4. **Motivation for EOF Analysis:**
   The rationale for employing EOF analysis to study soil moisture variability is not clearly justified. What specific insight does EOF provide in this context that other metrics might not? Please expand on the scientific motivation for this methodological choice.

   **Response:** To address the reviewer's request for a stronger justification of the EOF analysis, we have substantially revised Section 2.3 to clarify the rationale, methodological advantages, and contextual relevance of using EOFs to study soil moisture variability. The revised section now emphasizes that EOF analysis, implemented via Singular Value Decomposition (SVD), is particularly suited to extracting dominant spatiotemporal patterns from high-dimensional soil moisture datasets, enabling robust comparisons across experiments and observational benchmarks. We explicitly link this technique to our study's objective of evaluating how soil hydraulic parameterizations influence moisture dynamics across diverse hydroclimatic regions in CONUS. The revised text also highlights EOFs' utility in diagnosing interactions with large-scale climate drivers (e.g., ENSO, PDO), their role in identifying parameter-sensitive regions, and their advantage over simpler metrics such as RMSE or bias. Furthermore, we acknowledge the method's limitations, such as its reliance on orthogonality, and describe how we address these using supplementary diagnostics (e.g., Taylor diagrams, Euclidean distance). Lastly, we clarify that all EOF analyses were conducted using the open-source eofs (Dawson, 2016) Python package, ensuring transparency and reproducibility.

5. **Conclusion Structure:**
   The manuscript introduces two central research questions related to the influence of Soil hydraulic parameters on spatial soil moisture patterns and their temporal evolution during climate extremes. However, the conclusion section does not clearly revisit or synthesize findings in response to these questions. I recommend revising the conclusion to directly address the key research objectives and summarize how the results support them.

   **Response**: To address the reviewer's concern regarding the structure and focus of the conclusion, we have significantly revised Section 4 to explicitly revisit and synthesize the study's two primary research questions: (1) how soil hydraulic parameters influence spatial soil moisture distributions, and (2) how these parameters affect temporal dynamics during climate extremes. The updated section now integrates findings from the EOF and principal component analyses to demonstrate how parameter choices influence both spatial gradients, particularly in regions like ENA and CNA, and

the temporal evolution of moisture anomalies associated with events such as ENSO phases. In addition, we have expanded the conclusion to include a targeted set of practical recommendations for improving soil moisture simulation in CLM5. These include refining parameter estimation using advanced pedotransfer functions and machine learning methods, leveraging high-resolution satellite and in situ datasets (e.g., SMAP), and conducting region-specific parameter calibrations. We also emphasize the importance of accounting for vegetation–soil moisture feedbacks and linking modeled variability to large-scale climate drivers, such as ENSO, to enhance model realism and forecasting capability. These additions directly align the conclusion with the study's original objectives while offering clear directions for future land surface model development and application.

6. **Sensitivity of Hydraulic Parameters:**
It would be valuable for the reader to understand which specific Soil hydraulic parameters (e.g., saturated hydraulic conductivity, porosity, van Genuchten parameters) are most influential in controlling soil moisture dynamics across the simulations. A sensitivity analysis or discussion on this point would enhance the study's relevance for land model parameterization efforts.

**Response:** We appreciate the reviewer's suggestion to include a sensitivity analysis of individual soil hydraulic parameters. We agree that understanding the relative influence of specific parameters such as saturated hydraulic conductivity, porosity, and van Genuchten coefficients would provide valuable insights for model parameterization. However, this level of diagnostic analysis is beyond the scope of the present study. Our analysis is based on pre-run CLM5 simulations using prescribed parameter sets from SP-MIP, and we did not have access to the exact individual parameter values used within the model configurations for each experiment. As such, we were unable to systematically perturb or isolate individual parameters for a formal sensitivity analysis.

Instead, the study adopts a comparative experimental design outlined in Section 2.2.1, where each simulation applies a distinct parameter set derived from known pedotransfer functions or soil texture classes (e.g., EXP1 with standardized values, EXP4a–4d with uniform soil textures). Through this approach, we evaluated the aggregate effects of different parameter configurations on soil moisture variability, which were further decomposed using EOF analysis. While we acknowledge that this limits our ability to attribute responses to specific parameter changes, the results nonetheless highlight the substantial impact of parameter-driven variability on both spatial and temporal soil moisture patterns, especially in hydroclimatically complex regions such as the Great Plains.

We view this as an important direction for future research and have added to the revised manuscript (Section 4, see below), recommending that subsequent studies conduct targeted sensitivity analyses using parameter perturbation techniques or machine learning frameworks to systematically rank the influence of individual hydraulic properties on land surface model outputs.

*"....While our comparative framework assessed the aggregate effects of parameter set differences, we did not perform a formal sensitivity analysis to isolate the influence of individual soil hydraulic properties (eg., saturated hydraulic conductivity, porosity, van Genuchten parameters), which remains an important area for future investigation "*

**Reference**

Giorgi, F. and Francisco, R.: Evaluating uncertainties in the prediction of regional climate change, Geophysical Research Letters, 27, 1295– 1298, 2000.

Dawson, A.: eofs: A library for EOF analysis of meteorological, oceanographic, and climate data, Journal of Open Research Software, 4, 2016.

---

## Author Comment (AC6)

**RESPONSE TO REVIEWER #2 FOR GEOSCIENTIFIC MODEL DEVELOPMENT:**
**MANUSCRIPT EGUSPHERE-2025-713**

We sincerely thank Reviewer #2 for their thorough and insightful feedback on our manuscript, "Soil Parameterization in Land Surface Models Drives Large Discrepancies in Soil Moisture Predictions Across Hydrologically Complex Regions of the Contiguous United States." Your comments have been invaluable in helping us identify areas for improvement, and we outline below how we will address each point in the revised manuscript to enhance its clarity, robustness, and alignment with the standards of the hydrology and land surface modeling communities. Our responses to the suggestions are detailed below (in blue).

The authors present a comprehensive and methodologically rigorous study examining the influence of soil hydraulic and textural parameters on soil moisture simulations in CLM5, using soil parameter sets from the Soil Parameter Intercomparison Project (SP-MIP). Model outputs are compared against the ERA5-Land dataset as a benchmark. The study utilizes various analytical approaches, including means and variability assessments (Figures 4 and 5), as well as Empirical Orthogonal Function (EOF) analysis to investigate dominant spatial patterns and variability in soil moisture across the CONUS region. The findings suggest that soil parameterization has a substantial impact on CLM5 simulations, with notable discrepancies from ERA5-Land, particularly in hydrologically complex regions such as the Great Plains. The default CLM5 setup captures mean climatological patterns reasonably well but tends to underestimate interannual and seasonal variability.

Overall, the manuscript is well-structured, and the English is of generally high quality. The authors have executed a wide array of experiments that meaningfully contribute to our understanding of soil parameter sensitivities in land surface modeling. However, several critical issues remain that merit further investigation before the manuscript is suitable for publication. I recommend major revisions to address the following points:

**1. Limited Benchmarking Against Reference Data:**

The exclusive use of ERA5-Land as a benchmark is insufficient. While the authors acknowledge some of ERA5-Land's limitations, it remains a reanalysis product with inherent model dependencies and does not assimilate in-situ soil moisture observations directly. Prior studies [Koster et al., 2009] have demonstrated that soil moisture estimates are highly model-dependent. The validity of conclusions based solely on a single reference dataset is therefore limited.

To strengthen the analysis, I strongly recommend incorporating additional observation-based datasets, such as GLEM v3 *[Martens et al., 2017]*, SMERGE *[Tobin et al., 2019]*, and MERRA 2 [Reichle et al., 2017]. Each offers distinct advantages—GLEAM and SMERGE incorporate satellite-based observations, whereas MERRA-2 is a reanalysis-based soil moisture data. A recent study *[Duan et al., 2025]* has shown ERA5-Land's underperformance compared to these alternatives for sub-seasonal to seasonal forecast validations.

**Response:** We thank the reviewer for their thoughtful feedback regarding the use of ERA5-Land as the sole reference dataset. We respectfully clarify that the primary objective of our study was not to perform a formal validation of CLM5, but rather to assess the model's sensitivity to variations in soil hydraulic parameterizations. Our focus was on understanding whether the range of variability generated by CLM5 under different parameter configurations "brackets" the variability observed in ERA5-Land. The statistical tools we employed, such as EOF analysis, Euclidean distance, and Taylor diagrams, are commonly associated with model validation, but in our case, they were used as diagnostic tools to evaluate whether ERA5-Land's spatiotemporal variability could be reproduced through parameter perturbations alone.

We found that CLM5's parameter-driven variability consistently underestimated the amplitude of soil moisture variability compared to ERA5-Land, suggesting that the issue is not merely one of calibration but may reflect deeper structural limitations in how soil hydraulic properties are represented in the model. In this context, ERA5-Land serves as a physically consistent and spatially complete reference suitable for assessing relative variability patterns, rather than a ground-truth validation dataset. Although ERA5-Land does not assimilate in-situ soil moisture measurements (Muñoz-Sabater et al., 2021), its coherence and compatibility with CLM5's spatial and temporal scale make it well suited to this comparative framework.

We agree that incorporating additional observation-based datasets—such as GLEAM, SMERGE, or MERRA-2 could greatly benefit future studies aimed at model calibration. However, our current study is specifically focused on evaluating whether parameter variability alone can account for the structure of modeled soil moisture variability. Including multiple observational products at this stage would introduce additional complexity, potentially obscuring the sensitivity-based nature of our analysis. To address this, we have added clarifying text in Section 4 of the revised manuscript to better frame our work as a comparative sensitivity study. We also outline future directions that involve the integration of observational datasets for the purposes of model evaluation and calibration. Finally, we explicitly acknowledge the narrow scope of this study and emphasize the importance of using direct observational data in subsequent research focused on soil parameter calibration within global land models.

**2. Underestimation of Interannual and Seasonal Variability**

Figures 5 and 7 clearly indicate that all CLM5 configurations substantially underestimate soil moisture variability relative to ERA5-Land. Notably, Figure 5 reveals a tight clustering of CLM5 experiments, suggesting low variability in contrast to the wider spread of ERA5-Land data. However, this important point is underexplored in the manuscript. For instance, lines 284–285 state: "Despite these discrepancies, … broad agreement," which downplays the observed discrepancies. This variability gap warrants a deeper investigation and further supports the need for multiple observational references (as per Comment 1).

**Response:** We appreciate the reviewer's careful observation regarding the underrepresentation of soil moisture variability in CLM5 simulations compared to ERA5-Land, as illustrated in Figures 5 and 7. We fully agree that this discrepancy merits deeper treatment in the manuscript, both to reflect the limitations of the CLM5 parameterizations and to reinforce the case for additional benchmarking with observation-based datasets. To address this, we have made the following changes in the revised manuscript:

- Interannual Variability (Section 3.2 – Interannual Soil Moisture Anomalies):
  We have added a new discussion to explicitly highlight the tight clustering of CLM5 experiments and their muted anomaly spread relative to ERA5-Land. We interpret this as a systemic underestimation of interannual variability by the model, likely driven by diffusive parameter settings or limited responsiveness to hydrological extremes. The sentence previously beginning with "Despite these discrepancies..." (lines 284–285 in the original version) has been revised to emphasize that while CLM5 and ERA5-Land align in anomaly phase, they diverge significantly in magnitude. We also added supporting literature (e.g., Muñoz-Sabater et al., 2021), noting ERA5-Land's enhanced responsiveness to meteorological forcing due to its reanalysis design.

- Seasonal Variability (Section 3.3 – Seasonal Variability of Soil Moisture):
  We have inserted a new paragraph discussing the systematic underestimation of seasonal amplitude in CLM5 simulations, especially during the spring-summer transition. This observation, now clearly stated and referenced (e.g., Stahl and McColl, 2022), complements the interannual findings by showing that modeled soil moisture dynamics are dampened not only year-to-year but also within seasonal cycles. We interpret this underestimation as a likely artifact of overly conservative or static soil hydraulic properties in the parameter sets.

These additions enhance the manuscript's clarity and transparency regarding model limitations. We believe the revised text now better contextualizes the role of parameter choices in limiting model variability and strengthens our case for including multiple observational references in future work. We thank the reviewer again for bringing this important issue to our attention.

**3. Neglect of Irrigation Effects:**

The authors identify significant differences between CLM5 and ERA5-Land EOF modes in agriculturally intensive areas, particularly the central U.S. (Figure 11b). These "hotspots" overlap spatially with known heavily irrigated regions, including the Ogallala Aquifer and Mississippi Valley [McDermid et al., 2023]. However, the manuscript does not clarify:

- Whether Irrigation was included in CLM5 simulations.

- How Irrigation may affect soil moisture in ERA5-Land.

This omission weakens the attribution of model-observation discrepancies solely to soil parameterization. Explicit discussion on irrigation modeling and its inclusion or exclusion is essential to substantiate the attribution claims made (e.g., Lines 14–15).

**Response:** We thank the reviewer for this important comment regarding irrigation and its potential influence on the soil moisture discrepancies observed between CLM5 and ERA5-Land, particularly in the agriculturally intensive regions of the central U.S. We clarify that irrigation was not included in any of the CLM5 simulations used in this study. All experiments were conducted under naturalized conditions to isolate the influence of soil hydraulic parameterizations without additional confounding from anthropogenic water inputs. This decision aligns with our study's objective, which is to examine the intra-model sensitivity of CLM5 to soil parameter settings under consistent and climatically driven boundary conditions. We now explicitly state this in Section 2.2.1 (Experimental Designs):

*"....Importantly, irrigation processes were not represented in any of the CLM5 simulations, as all experiments were conducted under naturalized conditions to isolate the influence of soil hydraulic parameterizations without additional anthropogenic water inputs."*

Regarding ERA5-Land, we also confirm that irrigation is not represented in the ERA5-Land land surface simulations. The H-TESSEL model used to produce ERA5-Land does not include irrigation schemes or anthropogenic water management processes, as noted in previous studies (Wipfler et al., 2011; Lavers et al., 2022; Tang and McColl, 2023). We have incorporated this clarification into the revised Section 2.2.2 (Reference Dataset), noting that while ERA5-Land is a high-quality benchmark for pattern-oriented analysis, both CLM5 and ERA5-Land effectively simulate non-irrigated soil moisture dynamics. Therefore, the attribution of observed differences in EOF structure over agricultural "hotspots" to parameterization effects remains valid within the assumptions of this framework.

That said, we agree that in heavily irrigated areas such as the Ogallala Aquifer region or parts of the Mississippi Valley, the exclusion of irrigation from both datasets may limit interpretability, as true soil moisture conditions in these areas are influenced by human water use. We have included a statement to this effect in the revised Discussion, emphasizing that future work should incorporate irrigation modeling or use observation-driven products (e.g., SMERGE, GLEAM) that account for such influences, especially in agriculturally dominated landscapes. We appreciate the reviewer's guidance on this issue and believe the revised manuscript now provides a clearer and more transparent account of irrigation-related limitations and assumptions.

**4. Initial Conditions Not Explained:**

The setup of initial conditions in the model simulations remains unclear. Since each experiment involves different soil parameter settings, it is essential that the model reaches equilibrium separately for each case [Kennedy et al., 2024]. Without spin-up or appropriate initialization, differences in the initial soil moisture state could propagate and bias the results.

Please clarify whether each experiment was initialized to equilibrium independently, and if so, provide methodological details.

**Response:** We thank the reviewer for highlighting the importance of proper initialization and the potential impact of non-equilibrated soil moisture conditions. In response to this comment, we have revised Section 2.2.1 (Experimental Designs) to explicitly describe the spin-up procedure used for all CLM5 simulations.

As now clarified in the manuscript, each experiment (EXP1, EXP3, EXP4a–4d) was independently initialized using the standard CLM5 spin-up protocol. This approach involves running the model through an accelerated decomposition (AD) mode followed by a normal mode with repeated cycling of GSWP3 meteorological forcing. The spin-up process was conducted until key state variables such as total water storage, soil carbon, and vegetation biomass reached quasi-equilibrium, ensuring that each simulation began from a stable baseline (Lawrence et al., 2019). Spin-up followed SP-MIP protocol guidelines to ensure equilibrium prior to the 1980 to 2010 simulation period (Gundmundsson and Cuntz, 2017).

These updates now make it clear that any differences observed between experiments are attributable to the imposed soil parameter configurations and not to transient effects or inconsistencies in the initial conditions. We appreciate the reviewer's suggestion, which helped improve the clarity and methodological rigor of the manuscript.

**Minor Comments:**

- Line 32: The phrase "such as artificial neural networks" requires a reference. A citation demonstrating the use of ANN as PTFs would be appropriate. - We have included two citations: da Silva et al. 2023 and Schaap et al. 1998, to support the use of neural network-based PTFs.

- Line 77: Parentheses are inconsistent in references; e.g., fix "Ji et al……, Zeng et al., (2021)" to consistent formatting. - We have corrected this.

- Line 132: Typo: "PFTS" should be corrected to "PTFs."

  We have corrected the typo to refer to pedotransfer functions accurately.

- Lines 160–179 (Section 2.3): Consider revising for clarity. There are some repetitions, e.g., "dominant variability modes and their temporal patterns"…" spatial and temporal patterns"…

  We have streamlined the text to eliminate redundancies and enhance the description of the EOF analysis methodology.

- Line 193: The term "demeaned" could be clarified or rephrased for general readability.

  We have rephrased the sentence to read "... where the mean at each grid point has been removed to highlight variability."

- Figure 5: Further explanation is needed regarding the variability difference between ERA5-Land and CLM5 simulations.

  **Response:** We thank the reviewer for pointing out the need to clarify the variability difference between ERA5-Land and CLM5 simulations. In the revised manuscript, we have significantly expanded the "Interannual Soil Moisture Anomalies" subsection to address this issue. We now quantify the amplitude discrepancy, noting that ERA5-Land anomalies reach up to $\pm$ 40k\ m², while CLM5 anomalies are generally confined to a $\pm$ 20 kg\m² range. This underestimation reflects a muted response in CLM5, which we attribute to structural limitations such as static soil parameterization, overly diffusive vertical redistribution, and the absence of data assimilation factors that have been shown to reduce variability in land surface models (Koster et al., 2009; Muñoz-Sabater et al., 2021). We further discuss how this dampened variability may impact key hydrological processes like evapotranspiration, runoff, and land–atmosphere coupling (Koster et al., 2004; Berg and Sheffield, 2018). In addition, the caption for Figure 5 has been revised to clearly state the contrast in anomaly range between CLM5 and ERA5-Land, helping guide the reader's interpretation. These revisions directly respond to the reviewer's concern and enhance both the clarity and analytical depth of the section.

- Lines 218–319: The statement about 5 cm in-situ sensors vs. ERA5-Land's 0–7 cm integration is not directly relevant when the analysis uses 0–1 m averaged soil moisture. Clarify or remove.

  We have removed the statement to focus on the relevant depth.

- Figure 7: Please explain the significant differences between EAR5-Land variability and CLM5 cluster, except for Exp4-a.

  **Response:** We thank the reviewer for this helpful observation. In response, we have revised the Seasonal Variability of Soil Moisture *s*ubsection to provide a more detailed analysis of the differences between ERA5-Land and the CLM5 simulations (EXP1–EXP3), and to highlight the unique behavior of EXP4a. Specifically, we now explain that EXP1, EXP2, and EXP3 form a tightly clustered group with relatively flattened seasonal cycles that substantially underestimate the amplitude of variability observed in ERA5-Land. These configurations fail to capture the sharper rise in spring and pronounced decline in late summer exhibited by the reanalysis data. In contrast, EXP4a stands out as it more closely aligns with ERA5-Land in both phase and amplitude during the active seasonal months, which we attribute to its loamy sand texture and low water retention capacity. These additions clarify the model–reanalysis discrepancy, reinforce the role of soil texture in amplifying seasonal dynamics, and directly address the reviewer's request. We also revised the Figure 7 caption to highlight these findings more clearly.

- Figures 9 and A2: Clarify whether maps show EOF loadings or correlation coefficients. If correlation coefficients are used, explain the meaning and implications, e.g., correlation with respect to what?

**Response:** We thank the reviewer for raising this important point regarding the interpretation of the spatial patterns in Figures 9 and A2. We confirm that both sets of maps depict correlation coefficients, not raw EOF loadings. Specifically, the values represent correlation coefficients between the time series of each grid point's soil moisture anomalies and the corresponding principal component (PC) time series associated with the EOF mode. To address this comment and improve clarity for the reader, we have made the following revisions in the manuscript:

- Section 3.4.2 (Spatial and Temporal Analysis of EOF Modes) now explicitly states that the figures show correlation maps, not EOF loadings. We describe that the correlation coefficient quantifies the strength and direction of association between local soil moisture anomalies and the temporal evolution of each mode. A new sentence has also been added to explain that positive correlations indicate regions that vary in phase with the PC time series, while negative correlations reflect anti-phase behavior, thus providing insight into the regional expression of each mode.
- The caption of Figure 9 has been revised to clarify that the shading indicates correlation coefficients, and we have explained the interpretation of positive and negative values in the context of EOF-PC relationships.
- The caption of Figure A2 has been similarly updated to ensure consistency and interpretive transparency. It clearly states that the maps show correlation coefficients and describes what those correlations represent in terms of spatial coherence with the temporal EOF modes.

We believe these changes address the reviewer's concerns and improve the manuscript's clarity regarding the meaning and implications of the spatial EOF visualizations. We appreciate the reviewer's suggestion, which has helped us strengthen both the technical accuracy and readability of the manuscript.

- Table 3: Please include a total cumulative variance explained by the first 10 EOF modes for each experiment.

We have updated Table 3 to include the total cumulative variance explained.

| EOF Mode | EXP1 %Expl. Var. | EXP2 %Expl. Var. | EXP3 %Expl. Var. | ERA5-Land %Expl. Var. |
|---|---|---|---|---|
| EOF-1 | 11.45 | 11.66 | 10.84 ↓[2] | 17.5 ↓[2] |
| EOF-2 | 10.40 | 10.60 | 9.85 ↑[1] | 8.48 ↓[3] |
| EOF-3 | 8.81 | 8.25 | 9.08 | 7.83 ↑[1] |
| EOF-4 | 5.69 | 5.83 | 5.73 | 5.75 |
| EOF-5 | 4.37 | 4.59 | 4.48 | 5.61 |
| EOF-6 | 3.49 | 3.56 | 3.48 | 3.64 |
| EOF-7 | 3.26 | 3.23 | 3.24 | 3.10 |
| EOF-8 | 2.51 | 2.53 | 2.63 | 2.86 |
| EOF-9 | 2.14 | 2.16 | 2.22 | 2.76 |
| EOF-10 | 1.96 | 1.99 | 1.95 | 2.22 |
| Total Cumm. %Expl. Var. | 54.07 | 54.4 | 53.49 | 59.77 |

**Reference**

Tang, L. I. and McColl, K. A.: An Observational, Irrigation-Sensitive Agricultural Drought Record from Weather Data, Journal of Hydrometeorology, 24, 2091–2103, 2023.

Wipfler, E., Metselaar, K., Van Dam, J., Feddes, R., Van Meijgaard, E., Van Ulft, L., van den Hurk, B., Zwart, S. J., and Bastiaanssen, W. G.: Seasonal evaluation of the land surface scheme HTESSEL against remote sensing derived energy fluxes of the Transdanubian region in Hungary, Hydrology and earth system sciences, 15, 1257–1271, 2011.

Lavers, D. A., Simmons, A., Vamborg, F., and Rodwell, M. J.: An evaluation of ERA5 precipitation for climate monitoring, Quarterly Journal of the Royal Meteorological Society, 148, 3152–3165, 2022.

Stahl, M. O. and McColl, K. A.: The seasonal cycle of surface soil moisture, Journal of climate, 35, 4997–5012, 2022. Sullivan, P. L., Billings, S. A., Hirmas, D., Li, L., Zhang, X., Ziegler, S., Murenbeeld, K., Ajami, H., Guthrie, A., Singha, K., et al.: Embracing the dynamic nature of soil structure: A paradigm illuminating the role of life in critical zones of the Anthropocene, Earth-Science Reviews, 225, 103 873, 2022.

Lal, P., Singh, G., Das, N. N., Colliander, A., and Entekhabi, D.: Assessment of ERA5-land volumetric soil water layer product using in situ and SMAP soil moisture observations, IEEE Geoscience and Remote Sensing Letters, 19, 1–5, 2022.

Martens, B., Miralles, D. G., Lievens, H., Van Der Schalie, R., De Jeu, R. A., Fernández-Prieto, D., Beck, H. E., Dorigo, W. A., and Verhoest, N. E.: GLEAM v3: Satellite-based land evaporation and root-zone soil moisture, Geoscientific Model Development, 10, 1903–1925, 2017.

Lesinger, K. and Tian, D.: Trends, variability, and drivers of flash droughts in the contiguous United States, Water Resources Research, 58, e2022WR032 186, 2022.

Orth, R. and Seneviratne, S. I.: Analysis of soil moisture memory from observations in Europe, Journal of Geophysical Research: Atmospheres, 117, 2012.

Muñoz-Sabater, J., Dutra, E., Agustí-Panareda, A., Albergel, C., Arduini, G., Balsamo, G., Boussetta, S., Choulga, M., Harrigan, S., Hersbach, H., et al.: ERA5-Land: A state-of-the-art global reanalysis dataset for land applications, Earth system science data, 13, 4349–4383, 2021.

Hannachi, A., Jolliffe, I. T., and Stephenson, D. B.: Empirical orthogonal functions and related techniques in atmospheric science: A review, International Journal of Climatology: A Journal of the Royal Meteorological Society, 27, 1119–1152, 2007.

Tobin, K. J., Crow, W. T., Dong, J., and Bennett, M. E.: Validation of a new root-zone soil moisture product: Soil MERGE, IEEE Journal of Selected Topics in Applied Earth Observations and Remote Sensing, 12, 3351–3365, 2019.

Lawrence, D. M., Fisher, R. A., Koven, C. D., Oleson, K. W., Swenson, S. C., Bonan, G., Collier, N., Ghimire, B., van Kampenhout, L., Kennedy, D., et al.: The Community Land Model version 5: Description of new features, benchmarking, and impact of forcing uncertainty, Journal of Advances in Modeling Earth Systems, 11, 4245–4287, 2019.

Gundmundsson, L. and Cuntz, M.: Soil Parameter Model Intercomparison Project (SP-MIP): Assessing the influence of soil parameters on the variability of Land Surface Models, 2017.

Schaap, M. G., Leij, F. J., and Van Genuchten, M. T.: Rosetta: A computer program for estimating soil hydraulic parameters with hierarchical pedotransfer functions, Journal of hydrology, 251, 163–176, 2001

Duan, Y., S. Kumar, M. Maruf, T. M. Kavoo, I. Rangwala, J. H. Richter, A. A. Glanville, T. King, M. Esit, and B. Raczka (2025), Enhancing sub-seasonal soil moisture forecasts through land initialization, npj Climate and Atmospheric Science, 8(1), 100.

Kennedy, D., K. Dagon, D. M. Lawrence, R. A. Fisher, B. M. Sanderson, N. Collier, F. M. Hoffman, C. D. Koven, S. Levis, and K. W. Oleson (2024), One-at-a-time parameter perturbation ensemble of the community land model, version 5.1, Authorea Preprints.

Koster, R. D., Z. Guo, R. Yang, P. A. Dirmeyer, K. Mitchell, and M. J. Puma (2009), On the nature of soil moisture in land surface models, J Climate, 22(16), 4322-4335.

McDermid, S., M. Nocco, P. Lawston-Parker, J. Keune, Y. Pokhrel, M. Jain, J. Jägermeyr, L. Brocca, C. Massari, and A. D. Jones (2023), Irrigation in the Earth system, Nature Reviews Earth & Environment, 1-19.

Reichle, R. H., C. S. Draper, Q. Liu, M. Girotto, S. P. P. Mahanama, R. D. Koster, and G. J. M. De Lannoy (2017), Assessment of MERRA-2 Land Surface Hydrology Estimates, J Climate, 30(8), 2937-2960, doi: 10.1175/Jcli-D-16-0720.1.

Berg, A. and Sheffield, J.: Soil moisture–evapotranspiration coupling in CMIP5 models: Relationship with simulated climate and projections, Journal of Climate, 31, 4865–4878, 2018.

da Silva, L. d. C. M., Amorim, R. S. S., Fernandes Filho, E. I., Bocuti, E. D., and da Silva, D. D.: Pedotransfer functions and machine learning: Advancements and challenges in tropical soils, Geoderma Regional, 35, e00 720, 2023.

---

## Author Response (AR2)

**RESPONSE TO TOPIC EDITOR FOR GEOSCIENTIFIC MODEL DEVELOPMENT: MANUSCRIPT EGUSPHERE-2025-713**

We thank the Topic Editor for the thorough evaluation, constructive guidance, and recognition of the manuscript's contribution when framed as an intra-model sensitivity analysis within CLM5. Below, we provide a point-by-point response to each editorial request. The corresponding revisions have been implemented throughout the manuscript, including the Abstract, Introduction, Methods, Results, figure captions, and Conclusions. Our responses to the suggestions are detailed below (in blue).

**1. ERA5-Land framing and terminology**

**Editor Comment:** Recast ERA5-Land as a model product used for pattern comparison, not as "observations" or ground truth; avoid "performance/validation" language.

**Response:** We thank the Topic Editor for this essential clarification. We have implemented the requested reframing throughout. We now describe ERA5-Land as a spatially complete, model-based reference used only for pattern consistency (not validation of absolute levels). We explicitly state that ERA5-Land does not assimilate land/soil-moisture observations, is not ground truth, and is not a validation target. Instances of "observational benchmark," "validation," "performance," or "observed" have been replaced with phrasing such as "similarity/differences relative to ERA5-Land patterns." We chose the term "model-based reference" rather than "benchmark" to emphasize that ERA5-Land is another model product, not a standard of truth. This more cautious terminology avoids overstating its authority while maintaining its role as a consistent point of comparison for spatiotemporal pattern analysis.

**2. Forcing/structure mismatch is stated prominently**

**Editor Comment:** Note the forcing mismatch (CLM5 forced by GSWP3 vs. ERA5-Land forced by ERA5) and structural differences (CLM5 vs. HTESSEL); differences should not be attributed to parameter effects alone.

**Response:** We thank the Topic Editor for highlighting this distinction. We have added an explicit statement in *Methods* and reiterated it in *Results* and *Conclusions/Limitations* that differences between CLM5 and ERA5-Land can arise from both forcing differences (GSWP3 vs. ERA5) and model structural contrasts (CLM5 vs. HTESSEL), in addition to parameter effects. We emphasize this to avoid over-attribution of discrepancies to soil parameterization alone and to ensure that model-to-model and forcing mismatches are clearly recognized as contributing factors.

**3. Study the positioning and placement of objectives**

**Editor Comment:** Present plainly as an intra-model sensitivity analysis within CLM5, not a validation exercise; move objectives into the Introduction.

**Response:** We thank the Topic Editor for this guidance. We now state explicitly in both the *Abstract* and *Introduction* that this work is an intra-model sensitivity analysis within CLM5, not a validation exercise. ERA5-Land is framed only as a model-based reference for assessing pattern similarity. The two study objectives have been moved from the *Conclusion and Recommendation* to the end of the *Introduction* and reworded to align with the Methods framing. This restructuring ensures that readers understand from the outset that our focus is on documenting how soil parameterizations propagate into simulated variability within CLM5, rather than on validation against external products.

**4. Results wording and figure captions**

**Editor Comment:** Where ERA5-Land had been used to assert 'better/worse performance' or 'validation,' rephrase as 'greater/lesser similarity to ERA5-Land patterns' or 'pattern differences.

**Response:** We thank the Topic Editor for the specific wording direction. We have systematically replaced "performance/validation" language with pattern-based phrasing across the Results and all figure captions, e.g., "greater/lesser similarity to ERA5-Land patterns," "pattern differences," or "pattern consistency."

5. Compact experiment summary table; clarify Exp. 1 vs. Exp. 4a–d; fix cross-references

**Editor Comment:** "Include a compact experiment summary table; make Exp. 1 (single uniform parameter set) vs Exp. 4a–d (separate uniform texture-class sets, run individually) unambiguous. Correct figure/table cross-references."

**Response:** We thank the Topic Editor for this practical suggestion. We have added a concise experiment summary table listing EXP1 and EXP4a–d, including soil input, parameter settings, and purposes for each experiment. We clarify that EXP1 uses a single uniform parameter set globally, whereas EXP4a–d are four separate globally uniform "design-soil" runs (loamy sand, loam, clay, silt), each executed individually. We also state plainly that the analysis utilizes root-zone (0–1m) soil moisture data from 1980 to 2010. All figure/table cross-references have been corrected.

**6. Irrigation: explicit statement and reader caution**

**Editor Comment:** State plainly that neither CLM5 nor ERA5-Land includes irrigation; caution readers regarding agricultural hotspots.

**Response:** We thank the Topic Editor for this important caveat. We now explicitly state that irrigation is not represented in our CLM5 simulations (rainfed/naturalized) and is not included in ERA5-Land. We caution readers not to over-interpret agricultural hotspots for this reason. This clarification appears where the datasets are introduced and is reiterated in the Results and Conclusions.

**7. Brief Limitations paragraph (scope and proportionality)**

**Editor Comment:** Include a short Limitations paragraph acknowledging model-to-model comparison, forcing mismatch, and structural differences; keep conclusions proportionate.

**Response:** We thank the Topic Editor for emphasizing proportional conclusions. We added a concise Limitations paragraph in the Conclusions that: (i) reiterates the model-to-model, pattern-based scope of all comparisons (ERA5-Land used only as a model product for pattern comparison, not ground truth), (ii) notes the forcing and structural mismatches (GSWP3/CLM5 vs. ERA5/HTESSEL), and (iii) flags that neither dataset includes irrigation, so agricultural hotspots should be interpreted cautiously. We also state that parameter effects cannot be fully separated from forcing/structural differences, and results should be interpreted accordingly.

**8. Positive core and disciplined claims**

**Editor Comment:** The value lies in documenting, within CLM5, how parameter datasets propagate into regional patterns/variability and where parameter uncertainty most strongly affects simulated soil moisture, provided the claims are disciplined.

**Response:** We thank the Topic Editor for articulating the paper's positive core. We revised the Abstract and Conclusions to foreground the CLM5-internal contribution: documenting how soil parameter choices propagate into regional patterns and variability, and identifying regions/seasons where parameter uncertainty most strongly affects simulated soil moisture. Claims are now explicitly discipline-appropriate and proportionate, aligned with the pattern-comparison framing.

**9. Reviewer-specific technical points**

**Editor comment:** Please also address the specific technical points raised by Reviewer #3.

**Response:** We thank the Topic Editor for this reminder. We have prepared a separate, point-by-point response to Reviewer #3, and all technical comments are addressed in detail within it. Corresponding manuscript edits are implemented and cross-referenced as needed.

0/0-------

**RESPONSE TO REVIEWER #3 FOR GEOSCIENTIFIC MODEL DEVELOPMENT: MANUSCRIPT EGUSPHERE-2025-713**

We thank Reviewer #3 for the careful reading and constructive suggestions, which have improved the clarity, proportionality, and utility of the manuscript.

1. Lines 125–131 — Introduce only datasets; move parameter derivation to Sec. 2.2.1

**Reviewer Comment:** Introduce datasets only in Data Description; explain parameter derivation later.

**Response:** We thank the reviewer for this request for clarification. Implemented. The Data Description now introduces only the datasets (SP-MIP/CLM5 output and ERA5-Land) with domain, resolution, and period. All parameter-derivation details and the experiment setup are consolidated in Sec. 2.2.1 "Experimental Designs."

2. Line 127 — Table reference

**Reviewer Comment:** Should this refer to Table 2?

**Response:** Yes, the Brooks–Corey (Clapp–Hornberger) reference now points to Table 2.

3. Line 142 — Remove irrigated cropland

**Reviewer Comment:** Recommend removing irrigated cropland from analysis.

**Response:** We appreciate the reviewer's attention to this issue. We retain the full CONUS domain to preserve comparability across experiments and avoid introducing a non-standard mask; however, we now (i) state explicitly that irrigation is not represented in our CLM5 configuration and not included in ERA5-Land, and (ii) caution readers against over-interpreting agricultural hotspots. We flag this as a limitation and a priority for future work (e.g., using irrigation-aware datasets/masks).

4. Line 155 — Add a summary table of all experiments

**Reviewer Comment:** Please add a table summarizing experiments, datasets, and purposes.

**Response:** Implemented. We added a concise experiment summary table (EXP1–EXP4a–d) listing for each experiment the soil input, parameter setting, purpose, and the analysis period (0–1 m root-zone soil moisture, 1980–2010)

5. Line 156 — Clarify Exp. 1 vs. Exp. 4

**Reviewer Comment:** Both have uniform parameters—what is the distinction?

**Response:** Implemented. EXP1 applies one single uniform parameter set globally; EXP4a–d comprise four separate globally uniform "design-soil" runs (loamy sand, loam, clay, silt), each executed individually.

6. Line 316 — Figure cross-reference

**Reviewer Comment:** Should this be Figure 4d?

**Response:** Corrected. The cross-reference now points to Figure 4d; nearby figure/table references were re-checked for consistency.

7. Line 379 — Conclusions given ERA5-Land biases

**Reviewer comment.** If ERA5-Land has documented biases, what meaningful conclusions can be drawn from CLM5–ERA5-Land comparisons?

**Response:** We thank the reviewer for prompting a clearer interpretive scope. We now explicitly frame all CLM5–ERA5-Land comparisons as pattern-based, rather than as validation of absolute levels. We state that ERA5-Land is a model product without land/soil-moisture assimilation, and that CLM5–ERA5-Land differences reflect forcing (GSWP3 vs ERA5) and structural contrasts (CLM5 vs HTESSEL) in addition to parameter effects. Thus, our conclusions concern pattern consistency and parameter-sensitive regions, rather than errors in magnitude. We note that future extensions will incorporate observational products (e.g., SMAP, GLEAM, SMERGE, MERRA-2) to enable targeted calibration and broader evaluation.